# FREQUENCY-GUIDED MASKING FOR ENHANCED VISION SELF-SUPERVISED LEARNING

**Amin Karimi Monsefi[§], Mengxi Zhou[§], Nastaran Karimi Monsefi[†]**
**Ser-Nam Lim[‡], Wei-Lun Chao[§], Rajiv Ramnath[§]**
{karimimonsefi.1, chao.209, ramnath.6}@osu.edu, sernam@ucf.edu
[§]The Ohio State University, [†]Hamedan University of Technology, [‡]University of Central Florida

## ABSTRACT

We present a novel *frequency-based* Self-Supervised Learning (SSL) approach that significantly enhances its efficacy for pre-training. Prior work in this direction masks out pre-defined frequencies in the input image and employs a reconstruction loss to pre-train the model. While achieving promising results, such an implementation has two fundamental limitations as identified in our paper. First, using pre-defined frequencies overlooks the variability of image frequency responses. Second, pre-trained with frequency-filtered images, the resulting model needs relatively more data to adapt to naturally looking images during fine-tuning. To address these drawbacks, we propose **FO**urier transform compression with se**L**f-**K**nowledge distillation (**FOLK**), integrating two dedicated ideas. First, inspired by image compression, we adaptively select the masked-out frequencies based on image frequency responses, creating more suitable SSL tasks for pre-training. Second, we employ a two-branch framework empowered by knowledge distillation, enabling the model to take both the filtered and original images as input, largely reducing the burden of downstream tasks. Our experimental results demonstrate the effectiveness of **FOLK** in achieving competitive performance to many state-of-the-art SSL methods across various downstream tasks, including image classification, few-shot learning, and semantic segmentation. https://github.com/aminK8/FOLK

## 1 INTRODUCTION

In recent years, Self-Supervised Learning (SSL) has gained considerable interest in the context of visual pre-training. This interest stems from its prominent capability of extracting meaningful visual representations from the vast expanse of readily available, unlabeled images without the need for costly manual labeling (Ben-Shaul et al., 2024; Su et al., 2024; Almalki & Latecki, 2024). Key to this advancement is several pre-training methods established with different pretext tasks, including multi-view contrastive learning (Oord et al., 2018; Chen et al., 2020b; Tian et al., 2020b; He et al., 2020), Masked Image Modeling (MIM) (Bao et al., 2021; He et al., 2022a; Xie et al., 2022; Monsefi et al., 2024a; Oquab et al., 2024), Masked Frequency Modeling (MFM) (Xie et al., 2023; Liu et al., 2023; Zheng et al., 2024), and self-supervised Knowledge Distillation (KD) (Kakogeorgiou et al., 2022; Zhou et al., 2022; Chen et al., 2020c; Chen & He, 2021; Caron et al., 2021). In the recent popular approach of Masked Image Modeling (MIM), a key strategy involves masking portions of an image and then tasking models with either reconstructing these hidden sections or generating feature representations for them (Bao et al., 2021; Xie et al., 2022; He et al., 2022a; Yi et al., 2023). Through this process, the model is encouraged to learn robust feature representations that capture the underlying structure between unmasked and masked image parts, thereby enhancing its understanding of image semantics.

Instead of masking in the spatial domain (MIM), Masked Frequency Modeling (MFM) (Xie et al., 2023) introduced a self-supervised approach that masks frequency components of the input image. Since high-level semantics and low-level details of an image can be separated into different frequency components (Oppenheim & Lim, 1981; Piotrowski & Campbell, 1982; Navard & Yilmaz, 2024), the frequency domain offers a more convenient avenue for revealing underlying image patterns. The pretext task in MFM is to predict the masked frequencies from the frequency-filtered

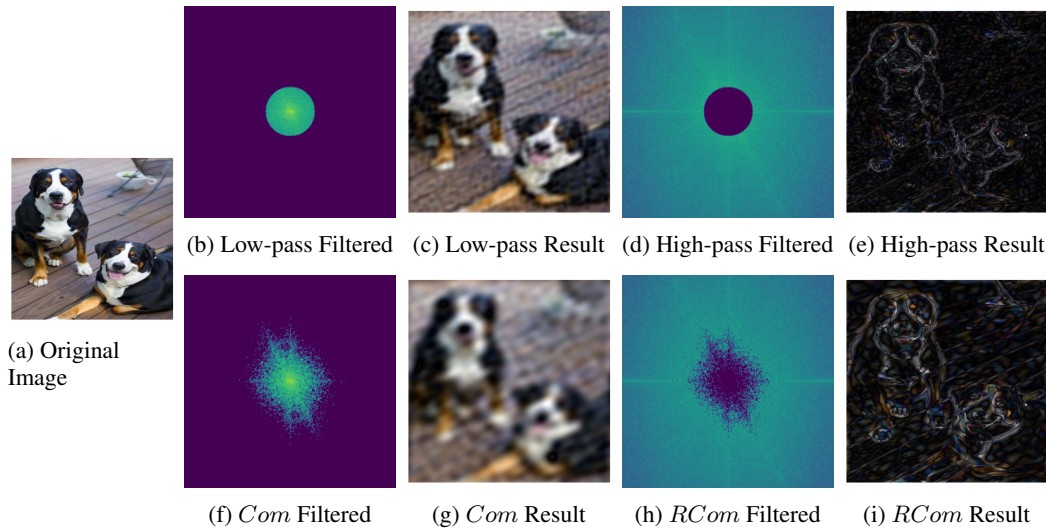

Figure 1: Detailed visualizations outlining our proposed masking approach compared to the low/high-pass filters used in the MFM method (Xie et al., 2023). Figures b, d, f, h here show the retained frequencies after applying the corresponding filter (zoom-in views on the shifted spectrum for better comparison). Figures c, e, g, i show the restored image from the retained frequencies. More examples can be found in Appendix D. All images used in this paper are from ImageNet (Deng et al., 2009).

image (see Section 3.1). It has two main advantages: first, it can help avoid issues encountered when analyzing raw pixel values in the spatial domain, such as spatial redundancy (Wang et al., 2020; Chen et al., 2023); second, unlike the patch-based masking used in MIM which often restricts the model to a Vision Transformer (ViT), MFM is suitable for both ViT and Convolutional Neural Network (CNN)-based models.

However, though MFM (Xie et al., 2023) has demonstrated promising results, it has two fundamental limitations. Firstly, MFM uses constant filters, controlled by a pre-defined hyperparameter *radius*, for masking in the frequency spectrum. This disregards the intrinsic structure specific to individual images, which leads to a less challenging task for reconstruction (see Figure 1b and Figure 1d). Secondly, in the pretext stage, MFM only shows frequency-masked images to the model without a mechanism to properly expose the raw information of the original images. This potentially restricts the pre-trained model's understanding of normal image distribution which further hampers MFM's training efficiency and model effectiveness, especially making it unsuitable in a few-shot learning scenario (see Section 4.2.2).

Our motivation is to achieve more effective vision pre-training through MFM, by addressing its aforementioned limitations. To this end, we propose a novel framework that integrates **FO**urier transform compression with se**Lf**-**K**nowledge distillation, termed as **FOLK**. Similar to MFM and dissimilar to MIM, FOLK applies masking to image frequency responses and embraces both ViT and CNN architectures. Furthermore, it resolves MFM's problems from two main perspectives.

Firstly, instead of adopting constant filters, we seek an improved masking scheme that considers each image's unique frequency responses, to improve the pre-training efficiency and model effectiveness. Inspired by the attention-based masking in MIM approaches such as AttMask (Kakogeorgiou et al., 2022), where the most critical parts of the image are hidden from the student model to create a more challenging pretext task, FOLK utilizes the Fourier transform compression (Pratt et al., 1969) concept to mask the most critical parts in image frequency responses. Two types of filters, the **Com** filter and its counterpart, the **RCom** filter, are designed to retain (or remove) the highest coefficient values within the frequency spectrum (see Fig. 1). In contrast to the constant filters used in MFM, these $Com$ and $RCom$ filters adaptively mask the frequencies that carry the essence (or finer details) of each individual image, creating greater variations across training samples. Hence, a

more challenging pretext task is presented to the model, enforcing its understanding of macro and micro visual cues uniquely held by each image.

Secondly, we question how to properly expose natural image information to the model during pre-training to enhance fine-tuning efficiency. To this end, FOLK incorporates a knowledge distillation strategy using a self-supervised teacher-student design. With the original image fed to the teacher model, and the frequency-masked image fed to the student, the student model learns not only to reconstruct the masked frequencies (as what MFM does), but also to reconstruct the original image's representation (generated by the teacher model) from the frequency-masked view of the same image. This multi-task teacher-student approach allows for model perception on both masked and original image realms, hence enhancing training stability and the pre-trained model's efficacy when applied to downstream tasks, as demonstrated in our experimental findings (Section 4.2).

To summarize, our contributions are threefold:

- We introduce a novel masking technique in masked frequency modeling with $Com$ and $RCom$ filters, which presents a meaningful and considerably more challenging pretext task for efficient SSL.

- We propose the FOLK framework, an innovative multi-task self-supervision methodology with self-knowledge distillation to allow for model perception on both frequency-masked images and original images in the pre-training stage.

- Through extensive experimentation, we demonstrate the efficacy of FOLK. Our findings indicate that FOLK performs on par or better than many state-of-the-art MIM and MFM techniques in various downstream tasks, including image classification, few-shot learning, and semantic segmentation.

## 2 RELATED WORK

### 2.1 SELF-SUPERVISED LEARNING

Self-supervised Learning (SSL) methods have been developed to exploit large-scale unlabeled data for learning discriminative representations, which can then benefit a variety of downstream tasks (Chong et al., 2023). Early SSL approaches rely on several pretext tasks, such as rotation prediction (Gidaris et al., 2018), jigsaw puzzle (Noroozi & Favaro, 2016), and colorization (Zhang et al., 2016). A branch of more recent studies follows a contrastive SSL paradigm (Chen et al., 2020b;c; Ci et al., 2022; Chen et al., 2020d; Tian et al., 2020b; Perera et al., 2024). SimCLR (Chen et al., 2020b;c; Monsefi et al., 2024b; Zhou et al., 2024) considers two augmentations of a given image as positives and the augmentations of all other images in the batch as negatives, on top of which a contrastive loss is utilized for model learning. MOCO (He et al., 2020; Chen et al., 2020d) maintains a dynamic dictionary of encoded representations with a momentum-updated encoder to generate consistent embeddings for contrastive learning. Instead of relying on negative samples and large training batches, BYOL (Grill et al., 2020) adopts a teacher-student framework that enforces consistency between representations of two augmented views of the same image generated by the two models. More recently, Correlational Image Modeling (CIM) (Li et al., 2023) operates by predicting correlation maps between randomly cropped image regions (exemplars) from a given image (context). It employs a bootstrap learning architecture with online and target encoders and a simple cross-attention mechanism to process the exemplars and context.

### 2.2 MASKED IMAGE MODELING

Masked Image Modeling (MIM) is an exciting approach to self-supervised visual learning. The central concept is to rebuild or reconstruct the hidden parts of images or to predict general characteristics like the image's category (Bao et al., 2021; Zhou et al., 2022; Chen et al., 2020a; Xie et al., 2022; Wei et al., 2022). It draws inspiration from the success of masked language modeling in natural language processing (Devlin et al., 2018; Liu et al., 2019), adapting the concept to the visual domain. Specifically, BEiT (Bao et al., 2021) utilizes a discrete pre-trained Variational AutoEncoder (VAE) to create discrete tokens for image patches, then it tasks the model with predicting such discrete tokens of masked patches in images. The iBOT method (Zhou et al., 2022) offers

improvements over BEiT by adopting a teacher-student self-distillation strategy where the teacher model simultaneously serves as an online tokenizer, instead of using a pre-trained discrete VAE. MAE (He et al., 2022a) takes a more aggressive masking strategy, with typically around 75% of patches being masked, and recovers the missing pixels using an autoencoder. To further investigate which parts of an image should be masked for efficient self-supervisory, AttMask (Kakogeorgiou et al., 2022) proposes the utilization of an attention map generated by a teacher model, and masks the highly attended patches from the image for the student model learning.

## 2.3 DISTILLATION-BASED MODELING

Knowledge Distillation (KD) in general endeavors to transfer knowledge from a complex, teacher model to its simpler, student counterpart (Buciluă et al., 2006; Hinton et al., 2015). This is often achieved by aligning the network logits (Hinton et al., 2015; Zhou et al., 2021) or intermediate representations (Romero et al., 2014), as well as designated statistics between teach and student models (Tian et al., 2020a; Ahn et al., 2019; He et al., 2022b; Chen et al., 2021). In the self-supervised domain, SEED (Fang et al., 2021) innovates by training a student encoder to mirror the similarity score distribution inferred by a larger, pre-trained teacher across a spectrum of instances. The EMA teacher, adopted by numerous SSL methodologies (Grill et al., 2020; He et al., 2020; Zhou et al., 2022; Caron et al., 2021), leverages the benefits of knowledge distillation to foster stabilized training and improved model efficacy. Our method extends the single model approach used by MFM (Xie et al., 2023) to a teacher-student distillation strategy for robust visual representation learning.

## 3 METHOD

### 3.1 PRELIMINARY AND BACKGROUND

In the domain of self-supervised learning for visual models, MFM (Xie et al., 2023) introduces a novel approach that diverges from traditional spatial domain masking strategies. By leveraging the frequency domain, which encapsulates both high-frequency details and low-frequency elements, MFM bases its learning process on the masking of frequency components and the prediction of the masked frequencies. More specifically, given a single-channel image[1] $x \in \mathbb{R}^{H \times W}$, the frequency representation is obtained via 2D Fast Fourier transform (FFT) $\mathcal{F}(x)$:

$$\mathcal{F}(x)(u,v) = \sum_{h=0}^{H-1} \sum_{w=0}^{W-1} x(h,w) e^{-i2\pi(\frac{uh}{H} + \frac{vw}{W})}, \tag{1}$$

where $x(h,w)$ represents the pixel value at the spatial coordinate $(h,w)$ on the image, while $\mathcal{F}(x)(u,v)$ denotes the complex frequency value at the coordinate $(u,v)$ on the spectrum. Here, $e$ is Euler's number, and $i$ is the imaginary unit.

To mask some frequencies from the spectrum and task a model with the reconstruction of these missing frequencies, a frequency-masked image $\tilde{x}$ is first obtained by:

$$\tilde{x} = \mathcal{F}^{-1}(\mathcal{F}(x) \odot M), \tag{2}$$

where $M \in \{0,1\}^{H \times W}$ is a mask, with 0 denoting corresponding frequencies being masked and 1 denoting frequencies being retained. $\mathcal{F}^{-1}$ denotes the inverse Fourier transform operation, and $\odot$ signifies element-wise multiplication. The learning objective of MFM, designed to minimize the discrepancy between the reconstructed and the original frequency components, can then be written as:

$$\mathcal{L}_{MFM} = \|(\mathcal{F}(x) - \mathcal{F}(g_\theta(\tilde{x}))) \odot (\mathbb{1} - M)\|_2, \tag{3}$$

---

[1]For RGB images, the procedure is applied to each channel independently.

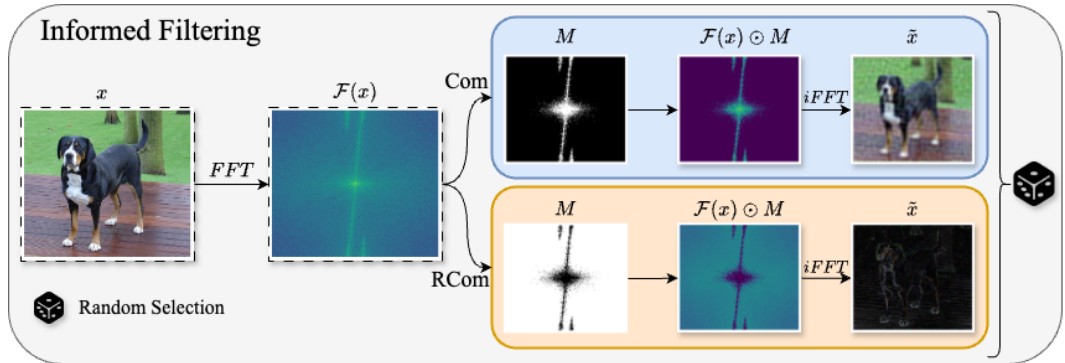

Figure 2: The frequency-based filtering process with our proposed informed filters, $Com$ and $RCom$. A portion of frequencies with the highest magnitude will be determined to create the masks. The process randomly returns the resulting image from the $Com$ or $RCom$ filter.

where $\mathcal{F}(x)$ is the frequency spectrum of the original image, $\mathcal{F}_r(\tilde{x})$ is the reconstructed spectrum using a neural network model $g_\theta$, parameterized by $\theta$. And $\mathbb{1} - M$ indicates that only the masked areas of the frequency spectrum are considered for loss.

## 3.2 FOLK FRAMEWORK

Before introducing our proposed method, we start by re-emphasizing the MFM method (Xie et al., 2023) limitations. Firstly, notice that, in Eq. 3, the MFM's loss depends on the filter $M$ applied to the spectrum. However, the low/high-pass filters used in MFM are simple, using a circular area with a fixed radius (see Fig. 1). This can lessen the difficulty of the frequency reconstruction task hence hampering the model learning. Another limitation of MFM is that the model only sees frequency-masked images in the pre-training stage, expressed by $g_\theta(\tilde{x})$ in Eq. 3. As a result, the pre-trained model may be relatively unfamiliar with natural images and requires more data during fine-tuning to adapt effectively (see Section 4.2.2).

To overcome these limitations and achieve effective masked frequency modeling, we propose the FOLK framework. Our key ideas involve the creation of informed frequency-based filters, $Com$ and $RCom$, as well as a self-distillation strategy based on a teacher-student design.

### 3.2.1 INFORMED FILTERS

Successful vision pre-training largely depends on the suitable and challenging enough pretext task presented to the model. Demonstrated by AttMask (Kakogeorgiou et al., 2022), masking the most-attended patches in images creates a more effective training scheme than random masking for MIM approaches. However, for MFM methods, this gap still exists as only constant masking/filters have been explored (Xie et al., 2023), which presumably presents a less challenging task in the pre-training.

To bridge this gap, we introduce two types of filters, $Com$ and $RCom$, for informed masking. Inspired by Fourier image compression techniques (Pratt et al., 1969), where the most significant frequencies (those with the highest magnitudes) that carry the bulk of an image's visual information are preserved while other frequencies are discarded for efficient storage or bandwidth usage, our $Com$ filters selectively retain these significant frequencies and discard the rest. This approach highlights the main semantics of an image for the model and necessitates the reconstruction of the less significant frequencies that correspond to finer details, such as edges. Conversely, $RCom$ filters remove the significant frequencies and require their reconstruction from the finer details preserved by the less significant frequencies. By applying both filter types during pre-training, the model is effectively trained to comprehend both macro and micro visual cues, thereby enhancing its generalizability and effectiveness in downstream tasks.

The generation of $Com$ and $RCom$ filters is illustrated in Fig. 2, with the pseudocode available in Appendix A.4. The input image is first converted to grayscale to ensure the creation of a single

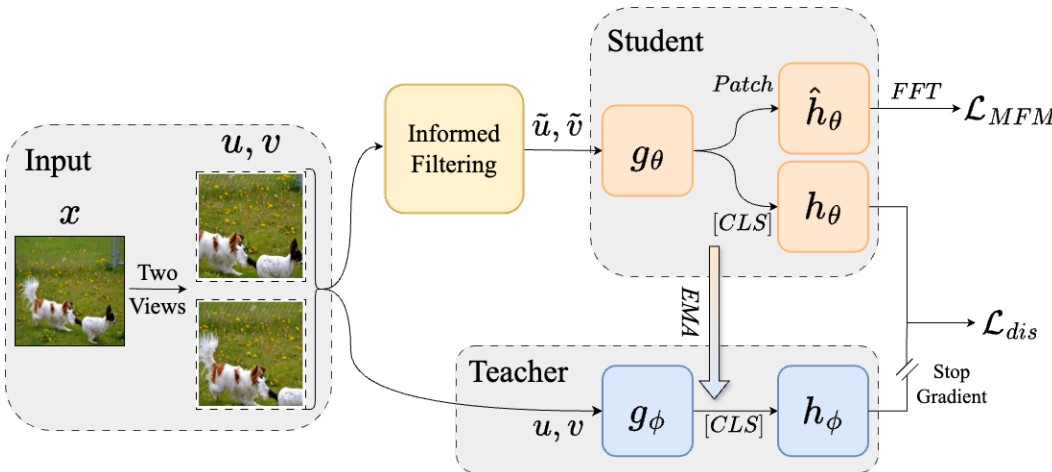

Figure 3: The Proposed FOLK Framework. Two views ($u$ and $v$) of the input image are processed through the informed filtering process, introduced in Figure 2. This yields two frequency-masked views ($\tilde{u}$ and $\tilde{v}$) which serve as the input to the student model. The student model is tasked with reconstructing the missing frequencies from the masked view $\tilde{u}$ (or $\tilde{v}$), as well as reconstructing the feature representation of the other original view $v$ (or $u$), generated by the teacher model, using the masked view $\tilde{u}$ (or $\tilde{v}$). The student model $g_\theta$ and the teacher model $g_\phi$, with their corresponding heads $h_\theta$ and $h_\phi$, have the same architecture but different parameters, except that the student has an additional MFM head $\hat{h}_\theta$. Only the student model (with its both heads) is updated through back-propagation, while the teacher parameters are periodically updated with an exponential moving average (EMA) of the corresponding student parameters.

common filter for all three RGB channels, as generating and applying filters separately to each channel can result in unnatural and corrupted visual information. The image is converted to a frequency spectrum using 2D FFT, after which the frequency components with the highest magnitudes are identified based on a threshold uniformly sampled from a predefined set of values ($[0.005, 0.01, 0.05]$ in our experiments, detailed in Appendix Section B.5.1). The informed filters are then created to retain ($Com$) or mask ($Rcom$) these frequencies. The two filters are randomly selected with an equal possibility (i.e. $50\%$) and applied to the spectrum (effects of different selection probabilities are provided in Appendix Section B.5.2). By applying inverse FFT to the filtered spectrum, we restore a frequency-masked image which will serve as an input to the model during pre-training. It is important to note that $Com$ and $RCom$ filters are uniquely generated based on individual images (examples are provided in Appendix D), which introduces more significant variation in training samples and presents a more challenging training task compared to using constant filters. In addition to comparing our approach with the low/high-passed filters used in MFM (Xie et al., 2023), we also conducted an ablation study using a set of random frequency filters to demonstrate the effectiveness of our proposed informed filters (refer to Appendix B.5.4).

### 3.2.2 MAKING BACKBONE FAMILIAR WITH NATURAL IMAGES

To further improve the training efficiency and model robustness in masked frequency modeling, we incorporate a self-distillation design (Grill et al., 2020; Caron et al., 2021; Tarvainen & Valpola, 2017) in our proposed approach. The original MFM (Xie et al., 2023) method only tasked the model with the reconstruction of missing frequencies from the frequency-masked view of the image. Such an approach potentially overlooked the data distribution of the original image space, as the model only sees frequency-masked images in pre-training, resulting in higher data demands to adapt to naturally looking images during fine-tuning. To resolve this issue, we propose to properly inject the original image information into the training process via a self-distillation technique (see Section 2.3 for more information about self-distillation). Here, we present our FOLK framework, detailed in Fig. 3. Note that FOLK does not require additional training stages for pre-processing, such as the offline tokenizer employed by BEiT (Bao et al., 2021).

FOLK starts by generating two views, $u$, and $v$, from an input image $x$, following the same data augmentations adopted by DINO (Caron et al., 2021). After applying 2D FFT to the view $u$ (or $v$), the $Com$ and $RCom$ filters are uniquely generated according to this view (detailed in Section 3.2.1 and Fig. 2). One filter is then randomly selected and applied to the frequency spectrum, and the retained frequency components are processed through inverse FFT to restore a frequency-masked view $\tilde{u}$ (or $\tilde{v}$), see Equation 2. The student model receives this frequency-masked view $\tilde{u}$ (or $\tilde{v}$) and predicts against two targets: reconstruction of the missing frequencies discarded by the filter, and reconstruction of the feature representation of the other original view $v$ (or $u$) generated by the teacher model.

Two different heads are appended to the student model, each facilitating one of the prediction tasks. The MFM head $\hat{h}_\theta$ serves to reconstruct the missing frequencies, with a single linear layer implementation following the original design proposed in MFM (Xie et al., 2023). The student head $h_\theta$ (and teacher head $h_\phi$), aiming at the reconstruction of the feature representation of the other original view, adopts a three-layer multi-layer perceptron (MLP) design. Moreover, the student head (and teacher head) is followed by a scaled softmax function to transform the outputs into probability distributions for the computation of a distillation loss (described below). More specifically,

$$P_s(x)^{(i)} = \frac{exp(f_\theta(x)^{(i)}/\tau_s)}{\sum_{k=1}^{K} exp(f_\theta(x)^{(k)}/\tau_s)}, \tag{4}$$

where $f_\theta(x) = h_\theta(g_\theta(x))$. $K$ is the output dimension of $h$, and $\tau_s > 0$ is a temperature parameter. A similar formula holds for the teacher counterpart with $P_t$ and $\tau_t$. To sharpen the output distributions—particularly for the teacher—we employ a scaled softmax function alongside the centering technique introduced in Caron et al. (2021). This combination helps prevent model collapse and reduces the dependency on large batch sizes during training. Detailed descriptions of the heads are provided in Appendix A.3.

The intended revealing of unmasked original image information is effectively achieved through FOLK's teacher-student design. During pre-training, the teacher model sees naturally looking images, which better align with those encountered during the fine-tuning stage, thereby enhancing fine-tuning efficiency, particularly in few-shot learning scenarios. On the other hand, the student model only observes masked views but is guided by the teacher model using the following distillation loss. Both the student and teacher models $g_\theta$ and $g_\phi$, with their heads $h_\theta$ and $h_\phi$, share the same architecture and initialization but have distinct parameters during training. Only the student model $g_\theta$ and its both heads $h_\theta$ and $\hat{h}_\theta$ are updated by the loss back-propagation, while the teacher parameters are periodically updated with an exponential moving average (EMA) of the corresponding student parameters.

A distillation loss is employed to enforce the less-informed student model to emulate the more knowledgeable teacher model who perceives the original views. For a single input image $x$, this loss can be written as:

$$\mathcal{L}_{\text{dis}} = -[P_t(u)\log\left(P_s(\tilde{v})\right) + P_t(v)\log\left(P_s(\tilde{u})\right)]. \tag{5}$$

where $u, v$ are two different views of $x$, and $\tilde{u}, \tilde{v}$ are the corresponding frequency-masked views. $P_s$ and $P_t$ are the student and teacher output probability distributions introduced in Eq. 4. The EMA update to the teacher is then ruled by $\phi \leftarrow \lambda\phi + (1 - \lambda)\theta$, ensuring a gradual integration of knowledge over time.

Note that, same as MFM (Xie et al., 2023), FOLK features compatibility with both ViT and CNN-based architectures. Illustrated in Fig. 3, when using a ViT-based model as the student (and teacher), the patch tokens out of the final encoder layer are fed into the MFM head, whereas the class token $[CLS]$ is directed to the student (and teacher) head. Conversely, when using a CNN-based model, the framework utilizes the final feature maps out of the CNN encoder as the input for the MFM head. An average-pooled feature map from the original feature maps will serve as the input to the student (and teacher) head. We provide experiments and results using a CNN model (i.e. ResNet-50 (He et al., 2016)) in Appendix B.2. Hence, FOLK facilitates a consistent approach across different architectural paradigms.

### 3.2.3 COMPREHENSIVE LOSS CALCULATION

Finally, a comprehensive loss can be derived by integrating the two primary loss components:

$$\mathcal{L}_{\text{tot}} = \alpha \cdot \mathcal{L}_{\text{dis}} + \mathcal{L}_{\text{MFM}}. \tag{6}$$

where a hyperparameter $\alpha$ controls the weights between two loss terms, which is set as 1 in our experiments, unless stated otherwise. Note that for a single input image, $\mathcal{L}_{\text{MFM}}$ takes an average over two terms for the two different views. This comprehensive loss facilitates the simultaneous model learning on the two tasks introduced above, the masked frequencies reconstruction (through $\mathcal{L}_{\text{MFM}}$) and original image feature reconstruction with self-distillation (through $\mathcal{L}_{\text{dis}}$). Furthermore, an ablation study for different choices of the hyperparameter $\alpha$ is provided in Section B.5.3.

### 3.2.4 COMPARISONS WITH OTHER SELF-DISTILLATION-BASED METHODS

Though FOLK deliberately leverages self-distillation to overcome the limitations of MFM — particularly its inability to expose the model to natural images — it is also important to clarify FOLK's distinctions from other self-distillation-based methods such as DINO (Caron et al., 2021), AttMask (Kakogeorgiou et al., 2022), and iBOT (Zhou et al., 2022). In contrast to the spatial domain augmentation paradigms adopted by these methods (e.g. random cropping and masking on the original images), FOLK performs augmentation in the frequency domain by masking with adaptive filters and presenting frequency-masked images to the student model to enhance its visual understandings. Note that the input to the teacher model is only spatially augmented (i.e. random cropping), thereby preserving the model's perception of natural images during pretraining. Moreover, FOLK incorporates a combination of both reconstructive and comparative losses, presenting rational yet more challenging pretraining tasks compared to approaches that rely solely on comparative losses, such as DINO and iBOT. Hence, we do not consider FOLK a close replication of previously presented methods. Please refer to Appendix Section C for a more detailed comparison.

## 4 EXPERIMENTS

In this section, we detail the experimental setup and evaluate our proposed FOLK framework on classification tasks using both full fine-tuning and few-shot learning approaches. Additional experiments, including semantic segmentation as a downstream task and ablation studies, are provided in Appendix B for further insights and comprehensive results.

### 4.1 SETUP

In this study, we employ three well-established model architectures as the foundation for our experiments: the ViT-Small (ViT-S/16), ViT-Base (ViT-B/16)(Dosovitskiy et al., 2021) and the ResNet-50 (He et al., 2016) (For ResNet-50 results see Appendix Section B.2). These models are chosen for their proven effectiveness and versatility, showing the power of our model with different types of model architecture. We adopt the ImageNet-1K training dataset (Deng et al., 2009) without labels for pre-training our self-supervised learning.

Our model's performance is evaluated across two critical areas: image classification and semantic segmentation. For image classification, we continue to leverage the ImageNet-1K dataset (Deng et al., 2009) to assess the generalizability and effectiveness of the learned features. In contrast, for semantic segmentation, we utilize the ADE20K dataset (Zhou et al., 2017), a standard benchmark in scene parsing and segmentation tasks. This bifurcated approach to evaluation ensures a thorough analysis of the models' capabilities in varied contexts. Our computational infrastructure supports these extensive experiments, consisting of four nodes, each of which has four NVIDIA A100 80GB GPUs, in total 16 GPUs. Please see Appendix A for implementation details.

| Method | Ref | Data | Epoch | Token | ViT-S | ViT-B |
|---|---|---|---|---|---|---|
| Scratch | (Touvron et al., 2021) | - | - | - | 79.9 | 81.8 |
| MAE | (He et al., 2022a) | INet-1K | 300 | ✓ | 80.6 | 82.9 |
| iBOT | (Zhou et al., 2022) | INet-1K | 1600 | ✓ | 81.1† | **84.0‡** |
| BEiT | (Bao et al., 2021) | INet-1K + DALL-E | 300 | ✓ | 81.3 | 82.9 |
| AttMask | (Kakogeorgiou et al., 2022) | INet-1K | 300 | ✓ | 81.3 | N/A |
| MoCo V3 | (Ci et al., 2022) | INet-1K | 600 | - | 81.4 | 83.2 |
| DINO | (Caron et al., 2021) | INet-1K | 1600 | - | 81.5 | 82.8 |
| MFM | (Xie et al., 2023) | INet-1K | 300 | - | 81.6 | 83.1 |
| MFM* | (Xie et al., 2023) | INet-1K | 300 | - | 81.2 | 82.9 |
| MFM + R/Com* | (Xie et al., 2023) | INet-1K | 300 | - | 81.4 | 83.2 |
| **FOLK** | Ours | INet-1K | 300 | - | 81.6 | 83.4 |
| **FOLK** | Ours | INet-1K | 800 | - | **82.1** | **84.0** |

Table 1: Top-1 results of fine-tuning self-supervised approaches utilizing ViT-S/16 and ViT-B/16 as encoders for INet-1K (ImageNet-1K). All recorded data were resized to images of size $224 \times 224$. Data means the pre-training dataset, and token means methods that need a masked token. *Our reproduced results with MFM official code through a pre-training phase of 300 epochs followed by 200 epochs of full fine-tuning. Also, MFM + R/Com means using the MFM approach with our proposed filters instead of its original low/high-pass filters. † Reported in AttMask paper with 300 pre-training epochs. ‡ iBOT ViT-B is pre-trained on 1600 epochs based on their report.

## 4.2 EXPERIMENTAL ANALYSIS

### 4.2.1 IMAGE CLASSIFICATION

In this section, we focus on the fine-tuning capabilities of different vision pre-training techniques using the ViT-S/16 and ViT-B/16 encoders on the ImageNet-1K dataset. The motivation behind this analysis stems from the need to understand how different SSL strategies, which range from traditional methods to novel approaches like the FOLK method introduced here, perform under uniform testing conditions with the well-established ImageNet benchmark dataset (Deng et al., 2009).

Table 1 presents the top-1 accuracy results of various SSL approaches fine-tuned on ImageNet-1K using both ViT-S/16 and ViT-B/16 encoders. Methods utilizing masked token strategies, such as MAE, BEiT, iBOT, and AttMask, consistently outperform the baseline, achieving up to $81.3\%$ with ViT-S and $84.0\%$ with ViT-B with huge number of pre-training epochs. These improvements highlight the effectiveness of masked token approaches in enhancing model performance, particularly when extensive pre-training epochs or additional data (e.g., DALL-E (Ramesh et al., 2021) for BEiT) are employed.

However, our proposed FOLK method demonstrates superior performance without relying on tokens or external datasets. With only 300 epochs of pre-training, FOLK achieves $81.6\%$ accuracy with ViT-S and $83.4\%$ with ViT-B, matching or, surpassing other methods that require more epochs or additional data. Extending the pre-training to 800 epochs, FOLK further improves the ViT-S and ViT-B accuracy to $82.1\%$ and $84.0\%$, surpassing all other methods in the ViT-S and ViT-B category under similar conditions or lower number of pre-training epochs. These results showcase FOLK's ability to enhance learning efficiency and model efficacy by integrating learning from dual inputs—filtered and original images—without the complexities of token masking or the need for external data sources.

### 4.2.2 FEW SHOT LEARNING

Our few-shot learning experiment's motivation lies in demonstrating the robustness and efficiency of our FOLK framework compared to other pre-training methodologies, especially in scenarios characterized by limited data availability. In this context, we particularly challenge the efficacy of the FOLK model against the MFM approach and others under the premise that showing original im-

ages (solving the second weakness of MFM by applying KD) significantly enhances performance on few-shot learning tasks.

In this experiment, we aim to highlight FOLK's superior adaptability and efficiency by fine-tuning pre-trained models using only $10\%$ of the ImageNet-1K dataset over 200 epochs. This setup allows us to critically assess the influence of learning rate adjustments on performance under sparse data conditions. We explore variations in base learning rates and warm-up periods using a cosine learning rate strategy for optimization (Gotmare et al., 2019).

| Method | BLR = 2e-4 WUp = 0 | BLR = 2e-4 WUp = 100 | BLR = 2e-3 WUp = 5 | AVG | MAX | Epoch |
|---|---|---|---|---|---|---|
| iBOT | 64.0 | 71.1 | 2.0 | 45.7 | 71.1 | 800 |
| AttMask | 69.8 | 71.0 | 31.3 | 57.4 | 71.0 | 300 |
| MFM | 57.7 | 58.5 | 41.9 | 52.7 | 58.5 | 300 |
| MFM + Com/RCom | 66.3 | 63.9 | 59.9 | 63.4 | 66.3 | 300 |
| **FOLK** | 71.2 | 68.1 | 62.2 | 67.2 | 71.2 | 300 |
| **FOLK** | 76.3 | 73.5 | 63.9 | 71.2 | 76.3 | 800 |
| **FOLK**$^{\ddagger}$ | **77.4** | **74.9** | **66.4** | **72.9** | **77.4** | 800 |

Table 2: Results of few-shot learning that fine-tunes the pre-trained ViT-S with different approaches for 200 epoch with $10\%$ of labeled data from ImageNet-1k. BLR: Base Learning Rate, means the peak value of the learning rate, and WUp: Warm Up, refers to the initial epochs during which the learning rate increases from 0 to the predefined BLR. After reaching the BLR, the learning rate then decreases according to a cosine function, from the BLR back down to 0. AVG: average. MAX: Maximum.$^{\ddagger}$ 1000 fine-tune epochs.

The results in Table 2 highlight FOLK consistently outperforms methods like iBOT, AttMask, MFM, and MFM with Com/RCom filters across different base learning rates (BLR) and warm-up (WUp) settings. For instance, with a BLR of $2 \times 10^{-4}$ and no warm-up (WUp=0), FOLK achieves a top-1 accuracy of $71.2\%$, surpassing iBOT's $64.0\%$, AttMask's $69.8\%$, MFM's $57.7\%$, and MFM with Com/RCom's $66.3\%$. When the pre-training epochs are increased to 800, FOLK's performance further improves, reaching $76.3\%$ accuracy under the same BLR and WUp settings, and even achieving $77.4\%$ when fine-tuned for 1000 epochs (denoted as FOLK$^{\ddagger}$). The average accuracy across the three learning rate settings for FOLK with 800 pre-training epochs is $71.2\%$, significantly higher than iBOT's $45.7\%$ with the same number of pre-training epochs. Notably, FOLK maintains robust performance even when the learning rate is less optimal, such as achieving $62.2\%$ accuracy with a higher BLR of $2 \times 10^{-3}$ and a short warm-up period (WUp=5), where other methods like iBOT drop drastically to $2.0\%$ accuracy. These results demonstrate that FOLK not only excels in leveraging limited labeled data but also exhibits resilience to variations in fine-tuning hyperparameters, underscoring its effectiveness and adaptability compared to other state-of-the-art methods in few-shot learning scenarios.

## 5 CONCLUSION

In this paper, we introduce the FOLK framework, a novel SSL method that addresses the limitations of previous frequency-based pre-training approaches. By integrating Fourier transform compression with self-knowledge distillation, FOLK adaptively selects frequencies for masking based on unique image responses, which allows the model to focus on more distinctive image features in the frequency domain, thereby enhancing the pre-training efficiency and model efficacy. Moreover, our dual-branch framework, which leverages both filtered and original images in pre-training, minimizes the adaptation requirements for natural-looking images in downstream tasks. Our experimental results demonstrate the effectiveness of FOLK, achieving competitive performance compared to many leading SSL methods. Notably, FOLK excels in tasks such as image classification, few-shot learning, and semantic segmentation, all while requiring fewer pre-training epochs.

ACKNOWLEDGMENTS

This project was made possible, in part, by support from the National Science Foundation grant number OAC-2018627 and from the Ohio Supercomputer Center.

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

APPENDIX

The following appendices provide in-depth information to supplement our main findings and methodologies presented in the paper. These appendices aim to enrich the reader's understanding by providing comprehensive insights and evidence supporting the robustness and versatility of our proposed method.

- Appendix A: Implementation details like pre-training, fine-tuning process, and projection heads architecture.
- Appendix B: Additional experimental results and analysis like semantic segmentation, few-shot learning, image classification for CNN-based model, and ablation study.
- Appendix C: Distinctiveness of FOLK with other SSL based model.
- Appendix D: Visual prediction results.

# A   IMPLEMENTATION DETAILS

Throughout our experimental investigations, we adopted the methodologies prescribed by the iBOT (Zhou et al., 2022) and MFM (Xie et al., 2023) frameworks. In a multi-GPU training circumstance, the learning rate adjustment is vital for optimizing the model's learning efficiency. The formulation used to compute the scaled learning rate is delineated below:

$$ScaledLR = BaseLR \cdot BatchSize \cdot \left(\frac{WorldSize}{512}\right) \tag{1}$$

In this context, $BaseLR$ signifies the optimal or peak learning rate identified for the model's training process. The term $BatchSize$ refers to the number of training examples processed simultaneously by each GPU. Lastly, $WorldSize$ denotes the total count of GPUs employed for parallel computation. The coefficient 512 in the denominator is a normalization factor, ensuring the scaled learning rate maintains an appropriate magnitude relative to the hardware configuration. We used the PyTorch Library (Paszke et al., 2019) for our code development.

## A.1   PRE-TRAIN STAGE

Our pre-training procedures largely align with those outlined in the BEiT (Bao et al., 2021) study, albeit with a few modifications. Specifically, our pre-training regime for the models incorporates simple yet effective data augmentation techniques, including random resized cropping to a resolution of $224 \times 224$ pixels and image flipping.

We employ the AdamW optimizer (Loshchilov & Hutter, 2019), with a pre-training duration set to 300 or 800 epochs, a batch size of 2048, 128 per GPU, and a peak learning rate of $1.2 \times 10^{-3}$. Additional parameters include a cosine decay learning rate schedule, 20 warmup epochs, and a specific setting for optimizer momentum ($\beta_1$, $\beta_2$ = 0.9, 0.95) (Chen et al., 2020a) with a weight decay of 0.05. Also, we used a value of 3.0 for gradient clipping to prevent the exploding gradient problem.

## A.2   FINE-TUNE STAGE

For the fine-tuning stage, we tried to keep most of the configuration of our FOLK the same as MFM (Xie et al., 2023) for a fair comparison.

### A.2.1   CLASSIFICATION TASK

We ran 200 epochs for fine-tuning the pre-trained model (i.e. ViT-S/16 or ViT-B/16) on ImageNet-1K for image classification, employing the $AdamW$ optimizer across all configurations with a weight decay of 0.05 and the optimizer momentum $\beta_1$, $\beta_2$ = 0.9, 0.999. Moreover, the approach includes a cosine decay learning rate schedule (Li & Arora, 2020), with a layer-wise learning rate

decay equal to $0.8$ (Bao et al., 2021; Clark et al., 2020). We also utilized advanced augmentation techniques such as Mixup (Zhang et al., 2018) and Cutmix (Yun et al., 2019), as well as label smoothing and random augmentation to further improve model robustness and generalization capability (Szegedy et al., 2016; Cubuk et al., 2020). The batch size is maintained at $2048$, with a peak learning rate set at $8 \times 10^{-3}$.

In contrast, the fine-tuning settings for the ResNet-50 model (He et al., 2016) generally follow the configurations suggested by (Wightman et al., 2021), with modifications to adopt the AdamW optimizer as recommended by (Fang et al., 2023). This includes a binary cross-entropy loss function (Zhang & Sabuncu, 2018) and adjustments to the learning rate scheduler. The weight decay is set to $0.02$, and the batch size is set to $2048$ to optimize performance. For ResNet-50, the fine-tuning epochs are specifically set to $300$, with distinct configurations for repeated and random augmentation, indicating a tailored approach to maximize the model's efficacy on the ImageNet-1K challenge.

### A.2.2 SEMANTIC SEGMENTATION TASK

We followed the pipeline demonstrated by the iBot (Zhou et al., 2022) paper for fine-tuning the pre-trained model for semantic segmentation using the $ADE20K$ dataset (Zhou et al., 2017). More specifically, we combined a pre-trained ViT-S/16 encoder with a UPerNet decoder (Xiao et al., 2018). The ViT-S/16 encoder extracts detailed features from images, while the UPerNet decoder specializes in semantic segmentation, translating these features into precise pixel-level classifications. This process employed the AdamW optimizer and fine-tuned for $160K$ iterations.

### A.3 PROJECTION HEAD

FOLK has three projection headers: one for frequency reconstruction (MFM Head, $\hat{h}_\theta$) and two for the student ($h_\theta$) and teacher ($h_\phi$) header, see Figure 3. We used a single linear layer for the frequency reconstruction head, similar to that in the MFM method (Xie et al., 2023). However, when it came to the distillation heads (student and teacher heads), we comprised a 3-layer multi-layer perceptron (MLP) with a hidden dimensionality of 2048. All layers were followed by a GELU activation except the final layer. We refrained from applying batch normalization (BN), as ViT architectures, unlike standard CNNs, typically eschew BN by default. Also, the output dimension adopts a dimensionality of 8192.

### A.4 PSEUDOCODE OF INFORMED FILTERS

Algorithm 1 presents the pseudocode for our proposed $Com$ and $RCom$ filters (denoted as $M$ in Eq. 2 and Eq. 3). Here, $S$ is the set of predefined values ($[0.005, 0.01, 0.05]$ in our experiments) from which a threshold is uniformly sampled to filter frequency components with the highest magnitudes. The parameter $com\_prob$ represents the probability of applying the $Com$ filter, set to $50\%$ in our experiments; accordingly, the $RCom$ filter is applied with probability $(1 - com\_prob)$. Ablation studies evaluating different choices of $S$ and $com\_prob$ are provided in Appendix Sections B.5.1 and Section B.5.2, respectively.

---

**Algorithm 1** $Com$ and $RCom$ Filter Generation

---

1: **procedure** FILTERGENERATION($image$, $S$, $com\_prob$)
2:     **if** $image$ is colored **then**
3:         Convert $image$ to grayscale.
4:     **end if**
5:     $spectrum \leftarrow$ apply 2D FFT to $image$ and retrieve the amplitude
6:     $compression\_rate \sim Uniform(S)$
7:     $threshold \leftarrow$ the $((1 - compression\_rate) \times 100)$th percentile of $spectrum$
8:     $Com\_filter \leftarrow (spectrum > threshold)$
9:     $RCom\_filter \leftarrow 1 - Com\_filter$
10:     Return $Com\_filter$ with a possibility of $com\_prob$, and $RCom\_filter$ with $(1 - com\_prob)$.
11: **end procedure**

---

## B    EXTRA EXPERIMENTS

### B.1    SEMANTIC SEGMENTATION

Semantic segmentation is a typical downstream task in the vision domain, where a classification needs to be performed on each pixel individually. We evaluated FOLK and compared it with several alternative SSL approaches on this task, using the ADE20K dataset (Zhou et al., 2017) and incorporating a task layer from UPerNet as described by (Xiao et al., 2018) to the SSL pre-trained encoder. The whole model was fine-tuned over 160k iterations, handling images at a resolution of $512 \times 512$, following the methodology established by iBOT (Zhou et al., 2022).

| Method | mIoU |
|---|---|
| Supervised Learning[•] (Zhou et al., 2022) | 44.5 |
| iBOT (Zhou et al., 2022) | 45.4 |
| iBOT+AttMask (Kakogeorgiou et al., 2022) | 45.3 |
| MFM[*] (Xie et al., 2023) | 44.9 |
| MFM + Com/RCom (Xie et al., 2023) | 45.1 |
| **FOLK**[†] | 45.3 |
| **FOLK**[‡] | **45.5** |

Table 3: The full fine-tuning ViT-S/16 model for semantic segmentation task with ADE20K dataset. [•] Supervised Learning result taken from iBOT paper. [*] We produced MFM results with their official code. FOLK was pre-trained with [†] 300 and [‡] 800 epochs.

Results of this fine-tuning effort for the semantic segmentation task on the ADE20K dataset are shown in Table 3. It presents a comparative analysis of different methodologies: Supervised Learning, iBOT, iBOT with Attention Mask (AttMask), MFM, and our FOLK model at different pre-training epochs (300 and 800 epochs). Notably, the FOLK model with 800 epochs of pre-training achieves the highest mIoU at 45.5, slightly surpassing the iBOT's 45.4 mIoU. This indicates a successful adaptation of the FOLK methodology, showing not only an improvement over the MFM results but also demonstrating that extended pre-training can lead to marginal yet significant performance gains.

### B.2    IMAGE CLASSIFICATION - CNN BASE MODAL

One limitation of many research studies, such as iBOT (Zhou et al., 2022) and AttMask (Kakogeorgiou et al., 2022), is that they are only compatible with a specific type of model architecture (e.g. ViTs), but not with others (e.g. CNNs). Our approach, like MFM (Xie et al., 2023), does not have this limitation and works with a wide range of encoder models.

Table 4 presents the ResNet-50 (He et al., 2016) (a CNN-based model) performance under the FOLK framework. The same FOLK pre-training strategies that have been applied to ViTs were seamlessly adopted here for CNNs. The only necessary modification to adopt CNNs is to alter the inputs to the heads: replacing patch tokens from ViTs with reshaped feature maps from CNNs, and replacing the $[CLS]$ token from ViTs with the average-pooled (and reshaped) feature map from CNNs. Our approach has demonstrated equal or superior performance to other methods with fewer epochs.

In the experiments above, we observed that FOLK did not lead to significant gains over MFM when using CNN-based models in the standard many-shot (full-data) setting. However, upon further investigation, we discovered that FOLK offers considerable advantages over MFM in few-shot learning scenarios, even with CNN architectures. We conducted additional experiments using a ResNet-50 (He et al., 2016) model as the backbone for both FOLK and MFM, and evaluated their performance on ImageNet-1k with limited labeled data (1% and 10% subsets). The experimental setup mirrored that of our ViT-based models to ensure a fair comparison.

As shown in Table 5, FOLK significantly outperforms MFM in the few-shot setting when using a CNN backbone. Specifically, FOLK achieves a top-1 accuracy of 71.1% with 10% labeled data and 46.2% with 1% labeled data, compared to MFM's 53.9% and 22.7%, respectively. These results highlight FOLK's robustness and effectiveness in scenarios with limited labeled data, making it a superior choice for applications where labeled data is scarce.

| Method | Ref | Epoch | Top-1 Acc |
|---|---|---|---|
| SimSiam | (Chen & He, 2021) | 400 | 79.1 |
| MoCo v2 | (Chen et al., 2020d) | 400 | 79.6 |
| SimCLR | (Chen et al., 2020b) | 800 | 79.9 |
| BYOL | (Grill et al., 2020) | 400 | 80.0 |
| SwAV | (Caron et al., 2020) | 600 | **80.1** |
| MFM | (Xie et al., 2023) | 300 | **80.1** |
| MFM + Com/RCom | (Xie et al., 2023) | 300 | **80.1** |
| **FOLK** | Ours | 300 | **80.1** |

Table 4: The top-1 full fine-tuning accuracy on ImageNet-1K for self-supervised models that utilize ResNet-50 as the encoder. This information compares our methods against others, with the results of these comparative methods being sourced from (Xie et al., 2023) and (Fang et al., 2023). The highest model performance is highlighted in bold, with the second highest being underscored.

| Model | Ref | Epoch | 10% | 1% |
|---|---|---|---|---|
| MFM | (Xie et al., 2023) | 300 | 53.9 | 22.7 |
| MFM + Com/RCom | (Xie et al., 2023) | 300 | 58.6 | 31.0 |
| **FOLK** | Ours | 300 | **71.1** | **46.2** |

Table 5: Few-Shot for ResNet-50 with 300 epochs pre-training.

The performance gains can be attributed to FOLK's self-distillation mechanism, which allows the student model to learn from the teacher model's representations of unmasked images. This mechanism provides additional guidance during pre-training, enabling the student model to better understand the underlying data distribution, even when the amount of labeled data for fine-tuning is limited. In contrast, MFM's reliance solely on reconstructing masked frequencies without exposure to unmasked images during pre-training may limit its effectiveness in low-data regimes.

In the full-data setting, the performance gap between FOLK and MFM using CNNs is less pronounced. This observation aligns with prior findings that CNNs, due to their inherent inductive biases and localized receptive fields, may not benefit as much from global context as ViT architectures do. Nonetheless, the substantial improvements observed in few-shot settings demonstrate FOLK's potential to enhance CNN-based models, especially when labeled data is limited.

These findings underscore the importance of evaluating self-supervised learning methods across different architectures and data availability scenarios. By effectively incorporating the original image information through self-distillation, FOLK enhances the pre-training process for CNNs, leading to better generalization in downstream tasks with scarce labeled data.

### B.3 ROBUSTNESS TO IMAGE NOISE AND ARTIFACTS

While our FOLK framework leverages the Fourier transform to create frequency-masked augmentations, concerns may arise regarding the potential sensitivity of the Fourier transform to image artifacts and noise. It is well-known that the Fourier transform can amplify the impact of noise due to its global nature (Oppenheim & Lim, 1981). However, our method uses the Fourier transform primarily to generate diverse and challenging masked inputs, and the actual model training occurs in the spatial domain.

To assess the robustness of our model's learned representations of image noise and artifacts, we conducted an experiment evaluating the impact of common types of image degradation on the model's performance in reconstructing masked frequencies. We randomly selected 100 images from the ImageNet-1k validation set and applied following types of degradation to each image:

- **Salt-and-Pepper Noise**: Randomly replaces pixels with maximum or minimum intensity values, simulating impulsive noise.

- **Gaussian Noise**: Adds zero-mean Gaussian noise with a standard deviation of $\sigma = 50$ to simulate sensor noise.

- **Brightness Variation**: Adjust the brightness by a random factor within $\pm 50\%$ to simulate different lighting conditions.

- **Contrast Variation**: Adjust the contrast by a random factor within $\pm 50\%$ to mimic variations in image quality.

For each original and degraded image, we applied the $Com$ or $RCom$ filters described in Section 3.2.1 to create frequency-masked versions. We then used our pre-trained FOLK model to predict the masked frequencies and calculated the MFM loss as defined in Equation 3.

| Noise Type | Without Noise | Salt-and-Pepper | Gaussian | Brightness | Contrast |
|---|---|---|---|---|---|
| **MFM Loss** | 0.179 | 0.181 | 0.182 | 0.186 | 0.189 |

Table 6: MFM loss values for original and degraded images. The slight increase in loss for noisy images indicates minimal sensitivity to noise. All the results of this table are collected by ViT-S/16.

Table 6 presents the average MFM loss values for the original images and images subjected to various types of degradations. The results indicate that these degradations cause only a slight increase in the MFM loss—approximately 0.002 to 0.01 higher than the loss for the original images. This marginal difference suggests that common image degradations do not significantly compromise our model's ability to reconstruct masked frequencies.

Several factors contribute to the robustness of our model to image artifacts and noise:

- **Spatial Domain Training**: Although we use the Fourier transform for masking, the input to the model remains in the spatial domain. This approach mitigates the potential amplification of noise in the frequency domain.

- **Randomized Masking Parameters**: The use of randomly sampled thresholds for filter creation (as discussed in Section 3.2.1) exposes the model to a wide range of frequency masking scenarios. This diversity encourages the model to focus on significant frequencies while reducing sensitivity to noise.

- **Data Augmentation**: Our training pipeline includes standard data augmentations such as random cropping and flipping. These augmentations help the model learn robust features that generalize well to variations in the input data.

The minimal impact of noise on the MFM loss indicates that our model effectively learns to reconstruct masked frequencies even when the input images contain artifacts. This robustness is crucial for practical applications where images may be degraded due to sensor imperfections or environmental conditions.

### B.4 FEW SHOT LEARNING

In addition to the primary few-shot learning results discussed in Section 4.2.2, Table 7 presents an extended evaluation of few-shot learning performance using a smaller set of labeled data. Various pre-trained models were fine-tuned using only $1\%$ of the ImageNet-1K dataset over 1000 epochs. This setup facilitates a detailed comparison of each model's ability to adapt to new data with minimal examples, highlighting a crucial aspect of model robustness and versatility. Furthermore, the evaluation considers three different settings for the base learning rate (LR) and warm-up periods, which are crucial hyperparameters in training deep learning models, especially under few-shot scenarios. The different configurations aim to assess each model's robustness across varying learning rate adaptation conditions.

The performance data presented in Table 7 underscores the robustness of the FOLK method across various learning rates and warm-up settings, evidenced by its superior average performance of $42.8\%$. This consistency indicates FOLK's inherent stability and adaptability in few-shot learning scenarios, distinguishing it from other models. Unlike iBOT and MFM, which exhibit fluctuating accuracies with changes in learning rates and warm-up periods, FOLK maintains a high level of performance. This suggests that FOLK is less sensitive to hyperparameter adjustments, thus requiring less fine-tuning to achieve good results. This characteristic is particularly significant in practical

| Method | BLR = 2e-4 WUp = 0 | BLR = 2e-4 WUp = 100 | BLR = 2e-3 WUp = 5 | AVG | MAX | Epoch |
|---|---|---|---|---|---|---|
| iBOT | 33.2 | 59.0 | 1.4 | 31.2 | 59.0 | 800 |
| AttMask | 50.4 | 59.1 | 3.2 | 37.6 | 59.1 | 300 |
| MFM | 26.9 | 31.6 | 6.3 | 21.6 | 31.6 | 300 |
| MFM + Com/RCom | 42.5 | 44.5 | 10.7 | 32.6 | 44.5 | 300 |
| **FOLK** | 51.7 | 56.1 | 20.5 | 42.8 | 56.1 | 300 |
| **FOLK** | **58.9** | **64.2** | **29.3** | **50.8** | **64.2** | 800 |

Table 7: Results of few-shot learning by fine-tuning pre-trained models for $1000$ epochs on $1\%$ of labeled ImageNet-1k. All models were sourced directly from their respective original repositories. All the results of this table are collected by ViT-S/16. BLR: Base Learning Rate, means the peak value of the learning rate, and WUp: Warm Up, refers to the initial epochs during which the learning rate increases from 0 to the predefined BLR. After reaching the BLR, the learning rate then decreases according to a cosine function from the BLR back down to 0. AVG: average. MAX: Maximum.

applications where extensive parameter tuning is inapplicable. By effectively integrating dual inputs—filtered and original images—FOLK enhances feature extraction and generalization capabilities, resulting in more reliable performance across various settings. This robustness, combined with a reduced dependency on precise parameter tuning, positions FOLK as an attractive option for tasks demanding high accuracy with minimal labeled data and limited pre-processing.

## B.5 ABLATION STUDY

### B.5.1 RATIONALE BEHIND THRESHOLD SELECTION

Our primary goal in selecting multiple threshold values was to introduce diversity in the frequency-masked input images during pre-training. By randomly sampling thresholds from a set (e.g., [0.005, 0.01, 0.05]), we ensure that the model is exposed to a variety of masking levels, which helps it learn more robust and generalized representations. The thresholds correspond to the proportion of frequency components retained or masked; for instance, a threshold of 0.005 means we retain (or mask) the top $0.5\%$ of the highest magnitude frequencies. By varying this threshold, we create different levels of difficulty for the reconstruction task, encouraging the model to learn both coarse and fine-grained features.

| Threshold(s) | CIFAR-10 | CIFAR-100 | ImageNet-1k |
|---|---|---|---|
| .05 | 98.8 | 90.2 | 81.1 |
| .01 | 99.0 | 90.7 | 81.4 |
| .005 | 99.0 | 90.6 | 81.3 |
| **.005,.01,.05** | **99.1** | **90.9** | **81.6** |
| **.002,.01,.07** | **99.1** | **90.9** | **81.6** |

Table 8: Evaluation of Com and RCom filters across benchmarks using varied threshold probabilities. Optimal results are achieved with thresholds [0.005, 0.01, 0.05], proving the method's effectiveness across diverse datasets. ViT-S/16 collects all the results of this table.

While the initial selection of these thresholds might seem arbitrary, our experiments indicate that the exact values are not highly sensitive parameters. To demonstrate this, we conducted additional experiments using different sets of thresholds and evaluated their impact on downstream task performance. We compared the original set of thresholds [0.005, 0.01, 0.05] with an alternative set [0.002, 0.01, 0.07]. The results, presented in Table 8, show that both sets yield similar performance across various datasets (CIFAR-10, CIFAR-100, and ImageNet-1k), and both outperform using a single threshold value.

These results suggest that the method's effectiveness is robust to the specific choice of thresholds as long as they are sufficiently separated to produce diverse frequency-masked images. The performance gains from using multiple thresholds indicate that the model benefits from the increased variability in the masking patterns. In practice, one can select a set of thresholds that produce visu-

ally distinct frequency-masked images without extensive tuning. We found that thresholds leading to the retention or masking of approximately $0.2\%$ to $7\%$ of the highest magnitude frequencies work well. Visual inspection of the resulting images can help in determining appropriate thresholds, as shown in Figures 13, 14, and 15 in the Appendix. By observing the images produced with different thresholds, practitioners can ensure that the selected thresholds provide a good balance between retaining important image structures and introducing sufficient masking to challenge the model during pre-training. The choice of multiple thresholds enhances the diversity of the training data without requiring extensive hyperparameter tuning. Our empirical results demonstrate that as long as the thresholds are spread out to produce varying levels of frequency masking, the model performance remains robust. This approach avoids the need for fine-grained tuning of threshold values and simplifies the practical application of our method.

In summary, while we initially selected the thresholds with the intention of creating diverse and challenging pre-training tasks, our experiments indicate that the exact values are not critical. The key is to have multiple thresholds that lead to different masking levels, and our findings show that the method is effective across various datasets with minimal sensitivity to these values.

### B.5.2 EFFECT OF FILTER SELECTION PROBABILITY

To assess the impact of different selection probabilities for the $Com$ and $RCom$ filters in our FOLK framework, we conducted an ablation study by varying the probability of selecting each filter during pre-training. While our default setting randomly selects between the two filters with equal probability $(50\%)$, this experiment aims to determine whether this choice is optimal and how sensitive our method is to this hyperparameter.

| Com Probability | RCom Probability | Accuracy |
| --- | --- | --- |
| 0.1 | 0.9 | 81.3 |
| 0.3 | 0.7 | 81.4 |
| **0.5** | **0.5** | **81.6** |
| 0.7 | 0.3 | 81.4 |
| 0.9 | 0.1 | 81.4 |

Table 9: The impact of different selection probabilities for the Com and RCom filters on model accuracy. A ViT-S model, 300 epochs of pre-training, and 200 epochs of fine-tuning on the ImageNet-1k dataset are adopted in these experiments.

The top-1 accuracy results on the ImageNet-1k validation set are presented in Table 9 We observe that selecting the filters with equal probability $(P_{\text{Com}} = 0.5)$ yields the highest accuracy of $81.6\%$. Slight deviations from this equal probability result in marginal decreases in performance, with accuracies ranging from $81.3\%$ to $81.4\%$. The performance differences are within $0.3\%$ of the highest accuracy, indicating that our method is not highly sensitive to the exact selection probability of the filters.

These results suggest that while an equal selection probability slightly outperforms other settings, the FOLK framework is robust to variations in the selection probability of the $Com$ and $RCom$ filters. The performance does not significantly degrade when the selection probabilities are skewed towards one filter over the other. This indicates that both filters contribute positively to the learning process, and the model benefits from the diversity introduced by both types of frequency masking.

### B.5.3 WEIGHT OPTIMIZATION

We explore the optimal weight values for $\mathcal{L}_{\text{tot}}$, introduced in Eq. 6. Table 10 provides an insight into how variations in the parameter $\alpha$ influence the Top 1 Accuracy of a ViT-S/16 model employing the FOLK methodology. As $\alpha$ is adjusted from 4 down to $0.05$, a clear trend is observed where the model's accuracy improves notably when $\alpha$ is reduced from 4 to 1, peaking at an accuracy of $81.6\%$ at $\alpha = 1$. This suggests that a lower $\alpha$ enhances the model's performance, potentially indicating an optimal configuration of the loss function $\mathcal{L}_{\text{tot}}$ at this point.

Further adjustments of $\alpha$ beyond this optimal point (decreasing it to $0.1$ and $0.05$) result in a slight decrease in accuracy to $81.4\%$, indicating a plateau. This stability around lower $\alpha$ values implies

that while $\alpha = 1$ is optimal, the model's performance does not degrade significantly with minor deviations from this value. The findings suggest that $\alpha$ critically influences the learning dynamics or loss function weighting, making its precise tuning essential for achieving the best performance from the ViT-S/16 model in the FOLK framework.

| Param | $\alpha = 4$ | $\alpha = 3$ | $\alpha = 2$ | $\alpha = 1$ | $\alpha = 0.1$ | $\alpha = 0.05$ |
|---|---|---|---|---|---|---|
| Acc | 80.7 | 80.8 | 81.2 | **81.6** | 81.4 | 81.4 |

Table 10: Effect of $\alpha$ values in $\mathcal{L}_{\text{tot}}$ on top 1 accuracy for a ViT-S/16 model using FOLK methodology.

### B.5.4 DIFFERENT FILTERS

Another part of our ablation study demonstrates the advantages of selective masking over random masking when applied to the frequency spectrum during model pre-training. The $Com$ and $RCom$ filters play a crucial role in optimizing self-supervised learning by leveraging the principles of image compression. The $Com$ filter focuses on major visual elements by preserving significant frequencies, thus compressing the image and emphasizing crucial features. Conversely, the $RCom$ filter retains less dominant frequencies to highlight finer details and textures, enhancing the model's sensitivity to subtle visual cues. This approach ensures that models are trained on both comprehensive and detailed representations, fostering adaptability and improved performance across diverse applications.

Additionally, our masking approach generates unique masks according to each image's frequency responses, thereby accounting for each image's distinctive features and semantics (see examples in the Tables 13, 14 and 15). This contrasts sharply with random masking. By targeting essential frequencies, selective masking ensures that the model adapts to recognize and prioritize these key signals, resulting in more robust and effective pre-training. Our findings indicate that this method significantly enhances the model's generalization capabilities and overall accuracy, confirming the efficacy of selective masking in the frequency domain for developing advanced predictive models.

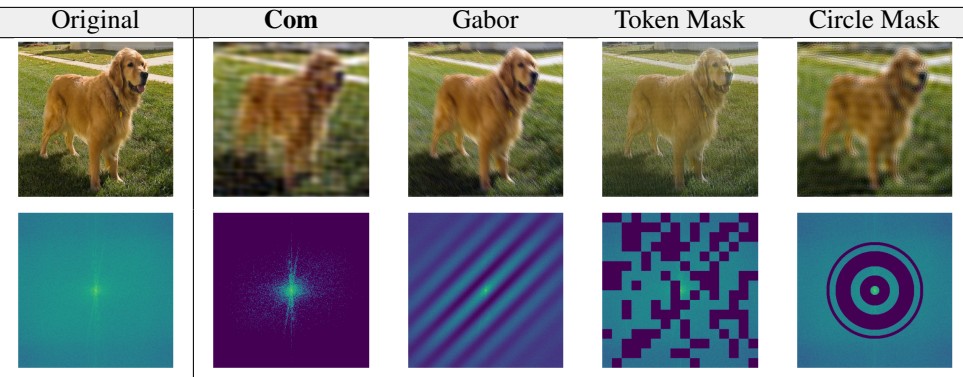

Figure 4: Visual comparison between $Com/RCom$ filters employed by FOLK and other random masking techniques.

Figure 4 illustrates the implementation of three supplementary random filtering techniques on the input frequencies and the effect of those filters compared with our $Com$ filter. The $Gabor$ filter that we directly applied in the frequency domain, (Kamarainen et al., 2006). In addition to this, $Token\ Mask$ is utilized, drawing inspiration from the $SimMIM$ approach (Xie et al., 2022), where random square regions are masked in the frequency domain to mimic missing information. Furthermore, we introduce the $Circle\ Mask$ strategy, which involves randomly masking circular areas at various distances from the center of the frequency spectrum. One of the significant advantages of our introduced filter ($Com/RCom$) is that it uses actual image information for masking, while other filters just randomly or constantly mask (like MFM) a portion of the frequency area without considering the structure and information of the image.

| Filters | Gabor | Token Mask | Circle Mask | $Com/RCom$ |
|---|---|---|---|---|
| **Acc** | 80.1 | 80.3 | 80.4 | **81.6** |

Table 11: Image classification results of utilizing different filters for masking in the FOLK framework. All the results of this table is collected by ViT-S/16.

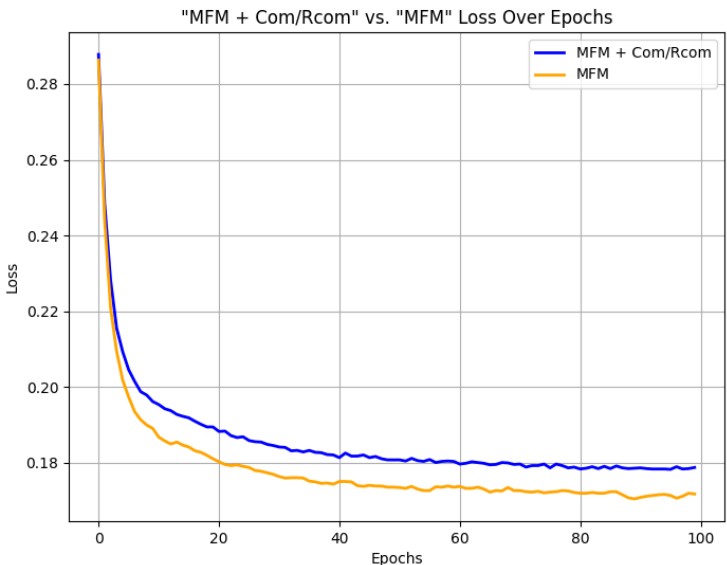

Figure 5: The loss progression of MFM using the original constant filters (denoted as "MFM") and MFM using the proposed $Com$ and $Rcom$ filters (denoted as "MFM + Com/Rcom") for the first 100 epochs during pre-training.

Table 11 illustrates the impact of various filtering techniques used for masking within the FOLK framework on model accuracy. The methods compared include Gabor filters, Token Mask, and Circle Mask filters. A clear pattern emerges from the data, with the $Com/RCom$ filters significantly outperforming the other techniques, achieving the highest accuracy at $81.6\%$. This indicates a superior efficacy of $Com/RCom$ filters in capturing and utilizing relevant image features for model training. The other filters—Gabor, Token, and Circle—also show competitive accuracies, but they are notably less effective than $Com/RCom$, with accuracies hovering around the $80\%$ mark. This suggests that the design and application of the $Com/RCom$ filters are better aligned with image intrinsic information, enhancing the model's ability to generalize from the training effectively.

### B.5.5 EFFECT OF INFORMED FILTERS ON PRE-TRAINING

To further illustrate the impact of the proposed $Com$ and $RCom$ filters on the pre-training process, we present in Fig. 5 a comparison of the loss progression of MFM when utilizing the original constant filters versus the $Com/RCom$ filters. Importantly, both experiments were conducted without the self-distillation design of FOLK, thereby isolating the analysis to the effect of the filters alone. As depicted in the figure, using the proposed $Com$ and $Rcom$ filters results in a higher MFM reconstruction loss (Eq. 3) throughout the pre-training process. This indicates that the model finds it more challenging to predict the masked frequencies produced by the $Com/RCom$ filters, suggesting that our filters are more effective in encouraging the model to learn richer representations by presenting more demanding reconstruction tasks.

## B.6 Efficiency Analysis

Our proposed enhancement involves augmenting the MFM model framework to accommodate dual inputs: both the original and the filtered images. By processing both types of inputs, the model gains a more comprehensive understanding of the data, which is particularly advantageous in scenarios requiring robust feature recognition, such as few-shot learning tasks. This approach aims to optimize the model's performance by leveraging the distinct characteristics captured in the filtered versus original images.

| Methods | **FOLK** | MFM | iBOT | AttMask* |
|---|---|---|---|---|
| MemGPU (GB) | 19.8 | **10.1** | 21.9 | 39.3* |
| Few-Shot Accuracy (%) | **67.2** | 52.7 | 45.7 | 57.4 |
| Classification Accuracy (%) | **81.6** | 81.4 | 81.1 | 81.3 |

Table 12: Comparison of GPU memory usage and model accuracy between FOLK and other methods. All methods are based on a ViT-S backbone. MemGPU is GPU memory with a batch size of 128. Few-Shot Accuracy is averaged from Table 2. Classification Accuracy comes from Table 1 in the main manuscript. *We could not run AttMask's official code with batch 128 with A100 80G and received a memory error. Hence, we ran it with batch 64.

To assess the effectiveness of our enhanced model, we provide memory usage on the GPU, and overall accuracy. Table 12 summarizes the performance metrics of various self-supervised learning methods, highlighting the impact of integrating dual inputs—original and filtered images—on model efficacy, particularly within the FOLK framework. Memory usage is a crucial consideration, as only GPUs with high capacity can run the model. This is noteworthy when compared to iBOT and AttMask which consume more memory[2], while our model requires less than 20GB of memory for the same batch size. The increased memory usage in models like FOLK and iBOT correlates with the advanced processing capabilities necessary for handling dual inputs. However, the evident gains in learning accuracy justify the resource allocation, making it a worthwhile trade-off for applications where high precision and robust feature recognition are critical, such as few-shot learning scenarios. FOLK achieves the highest few-shot learning accuracy at 67.2% and tops classification accuracy at 81.6%. This performance indicates that the model's ability to effectively utilize both filtered and original images significantly enhances its learning capabilities, particularly in tasks requiring robust feature recognition.

## C Comparison with DINO and Distinctiveness of FOLK

While our proposed FOLK framework and DINO (Caron et al., 2021) both employ self-distillation techniques, it is crucial to clarify the key differences between the two methods and highlight the unique contributions of FOLK.

### C.1 Use of Self-Distillation

Both FOLK and DINO utilize knowledge distillation in a self-supervised learning context involving a student-teacher architecture where the student model learns from the teacher model's output. However, knowledge distillation is a common technique in self-supervised learning (SSL) and has been adopted in several works such as BYOL (Grill et al., 2020), iBOT (Zhou et al., 2022), and AttMask (Kakogeorgiou et al., 2022). Therefore, using self-distillation alone does not define a method's identity or make it equivalent to DINO.

### C.2 Addressing Different Problems

FOLK integrates self-distillation to address a specific limitation in Masked Frequency Modeling (MFM) (Xie et al., 2023). Namely, the model lacks looking at the natural view of images and unmasked images during pre-training. This limitation can hinder the model's performance, especially in few-shot learning scenarios where labeled data is scarce. By incorporating self-distillation, FOLK

---

[2]AttMask cannot run with a batch size of 128 on an A100 80GB GPU, so we run it with a batch size of 64.

allows the student model, which only sees frequency-masked images, to learn from the teacher model's representations of unmasked images, thereby enhancing its ability to generalize to natural images.

### C.3 FREQUENCY-BASED MASKING AND FILTERS

A key distinction of FOLK is the introduction of informed frequency-based filters ($Com$ and $RCom$) for masking in the frequency domain. These filters are designed to adaptively mask significant frequencies based on each image's unique frequency components, creating a more challenging pretext task and enhancing the model's ability to learn both macro and micro visual cues. This approach fundamentally differs from DINO, which operates in the spatial domain.

### C.4 COMPUTATIONAL EFFICIENCY

FOLK is designed to be computationally efficient compared to DINO. Notably, FOLK requires fewer pre-training epochs to achieve competitive or superior performance (see Table 1). For instance, FOLK achieves strong results with 300 or 800 pre-training epochs, whereas DINO typically requires 800 to 1600 epochs to reach even lower performance levels (Caron et al., 2021).

Additionally, FOLK has lower GPU memory requirements. Using a batch size of 128, FOLK's GPU usage is approximately 19.8 GB (see B.6), while DINO's GPU usage is around 30.8 GB (based on reported usage in (Caron et al., 2021) with adjusted batch sizes). This efficiency is partly due to FOLK not requiring additional local views of images during training, unlike DINO.

### C.5 METHODOLOGICAL DIFFERENCES

While both methods use self-distillation, FOLK and DINO differ in their training methodologies:

- **Input Views**: FOLK employs frequency-masked views generated using $Com$ and $RCom$ filters, whereas DINO uses multiple spatial views, including global and local crops. FOLK does not need any local view.
- **Loss Functions**: FOLK combines a reconstruction loss (MFM loss) for masked frequency prediction with a distillation loss, enabling the model to learn both from reconstructing missing frequencies and from the teacher's representations. DINO relies solely on the distillation loss.
- **Exposure to Original Images**: In FOLK, the teacher model is exposed to unmasked images, and the student learns to reconstruct the teacher's representations from frequency-masked images. This design helps the student model adapt to natural images despite not seeing them directly during training.

## D  PREDICTION VISUALIZATION

This section presents visualizations of a series of images from the ImageNet-1K dataset, processed by our newly proposed techniques, namely the $Com$ and $RCom$ filters. Tables 13, 14, and 15 collectively demonstrate the impact of our proposed approach. Distinct from the MFM method, which employs static and conventional low/high-pass filters, $Com$ and $RCom$ filters are dynamic and tailored to each image. This adaptability allows the filters to change in response to an image's unique concept and structural characteristics, offering a more nuanced and effective processing method. In addition, Tables 13, 14, and 15 also show the reconstructed missing frequencies for each image, generated by the FOLK pre-trained model. The clear alignments between the model predicted and the ground-truth masked frequencies provide further evidence for the successful pre-training of FOLK and, therefore the improved model efficacy.

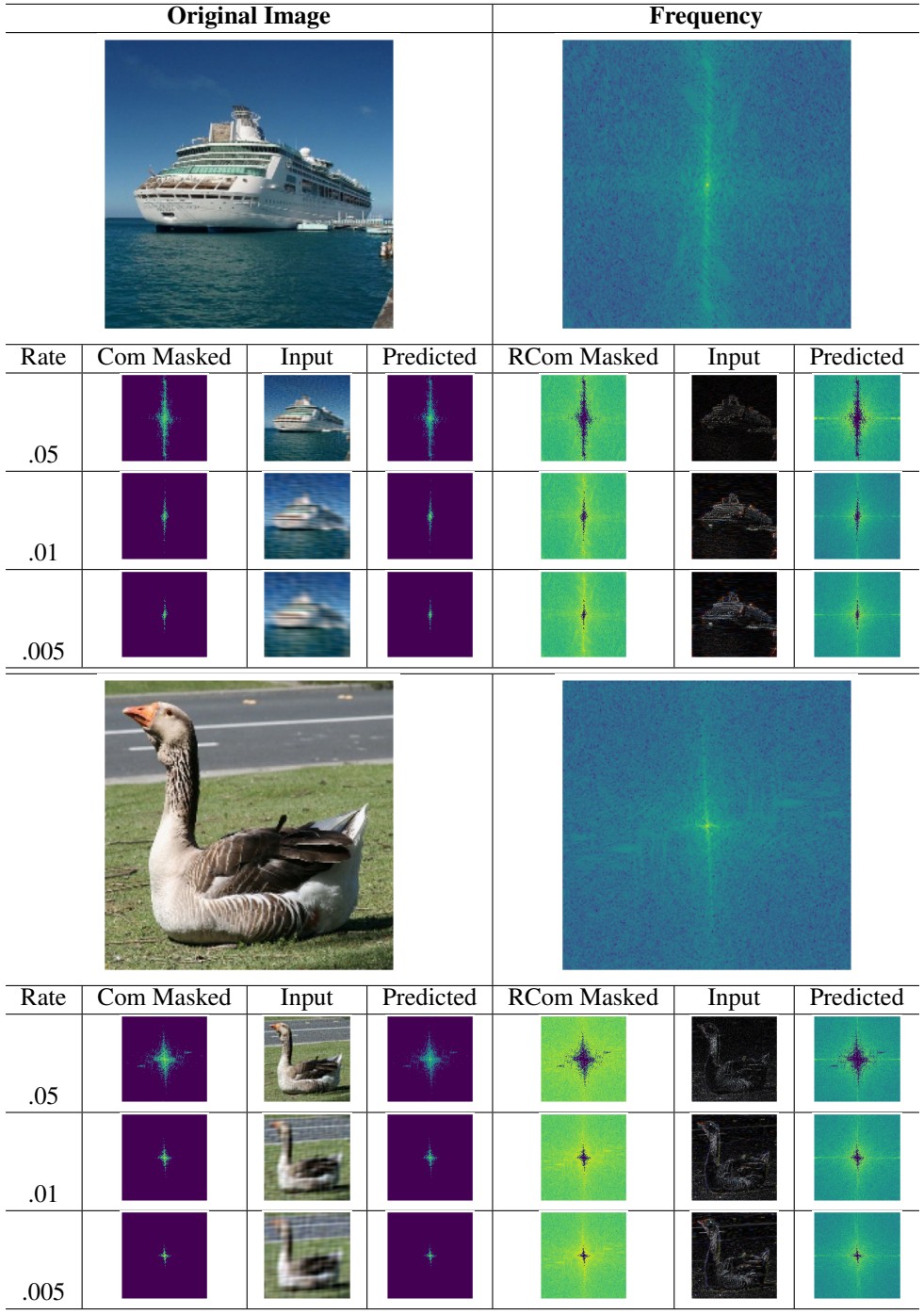

Table 13: The effect of applying $Com$ and $RCom$ filters to different images, and the predicted missing frequencies by the FOLK pre-trained model. Rate means the rate of compression (between 0 and 1).

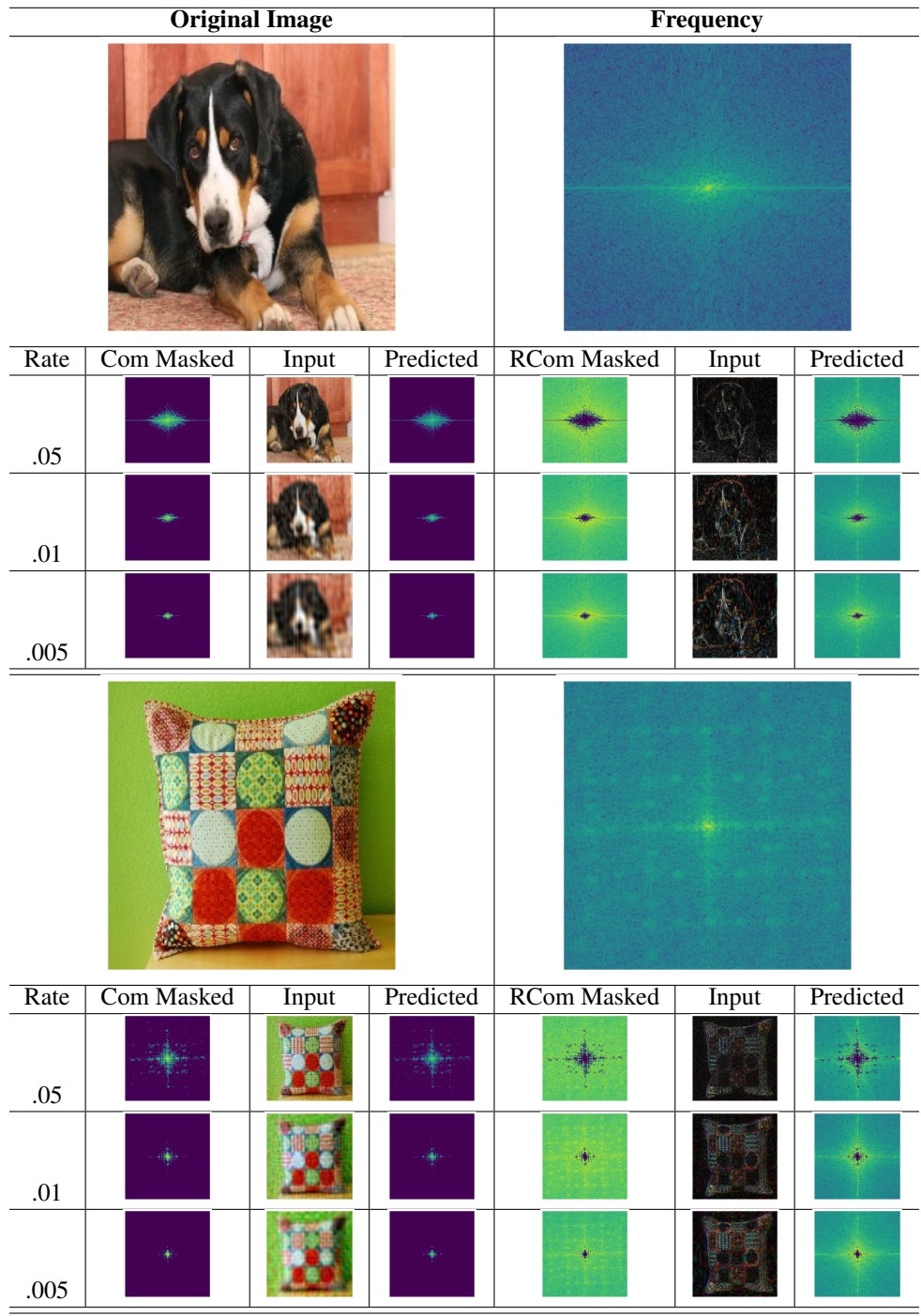

Table 14: The effect of applying $Com$ and $RCom$ filters to different images, and the predicted missing frequencies by the FOLK pre-trained model. Rate means the rate of compression (between 0 and 1).

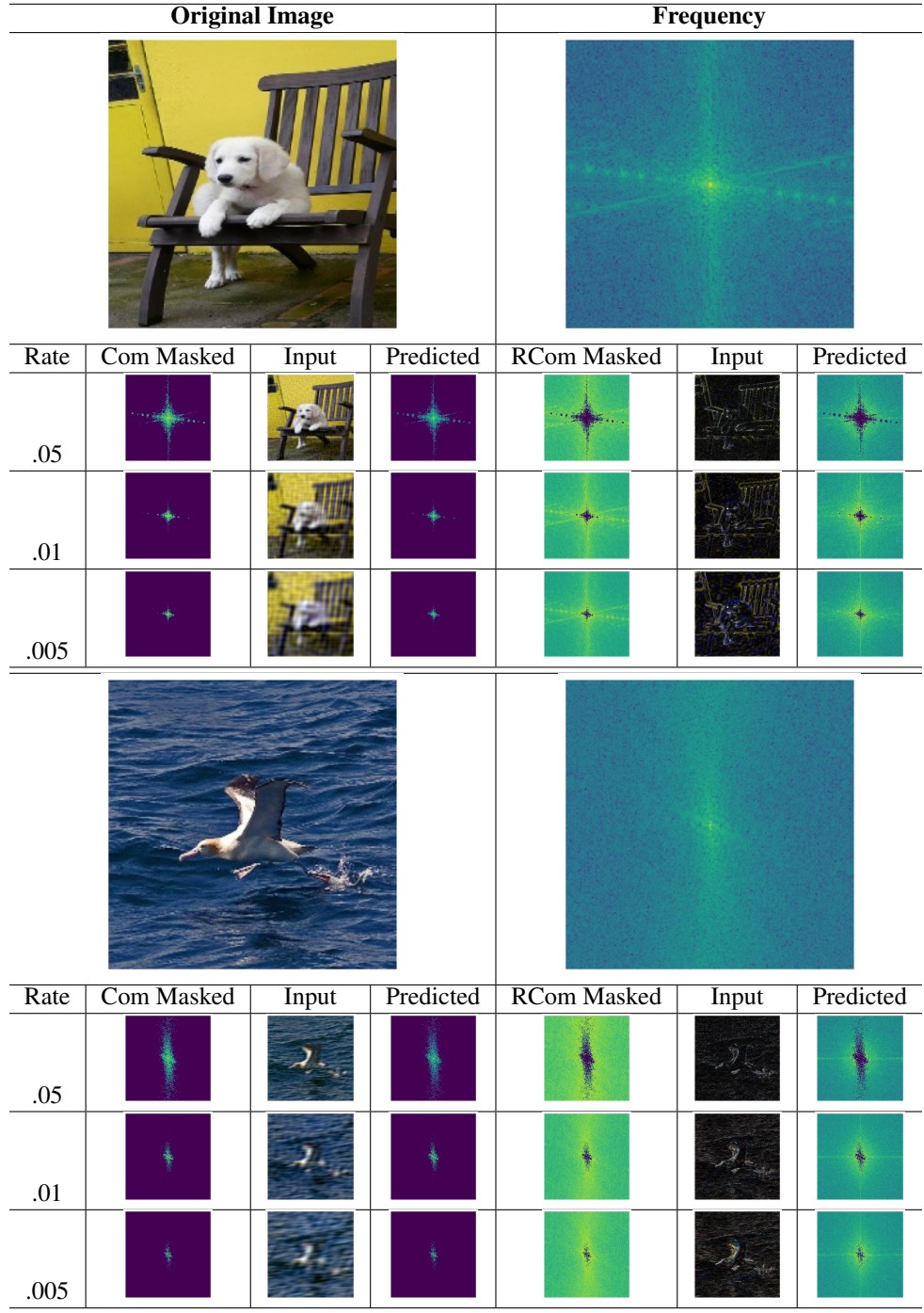

Table 15: The effect of applying $Com$ and $RCom$ filters to different images, and the predicted missing frequencies by the FOLK pre-trained model. Rate means the rate of compression (between 0 and 1).

