# OpenReview forum: "Frequency-Guided Masking for Enhanced Vision Self-Supervised Learning"
_ICLR.cc/2025/Conference — ICLR 2025 Poster_

### Official Review · Reviewer_6EN5 · 2024-11-03

**Soundness:** 3
**Presentation:** 3
**Contribution:** 3
**Rating:** 8
**Confidence:** 5

**Summary:**

The authors identify two fundamental limitations in the existing masked frequency modeling (MFM) paradigm: 1) constant filters overlook the variability of image frequency responses, and 2) no access to naturally looking images during pre-training requires more data to adapt to downstream tasks during fine-tuning. To address 1), the authors adaptively select the masked-out frequencies based on image frequency responses. To address 2), the authors employ a student-teacher framework via self-distillation. Experimental results on image classification, few-shot learning, and semantic segmentation demonstrate the effectiveness of the proposed method compared to the MFM baseline.

**Strengths:**

-	The proposed method is well motivated. The authors motivate the method by identifying two key limitations in MFM, and propose two interesting solutions to address these drawbacks.
-	The paper is generally well-written and easy to follow.
-	The authors provide a comprehensive analysis based on their method. The experiments are extensive and the results are promising, especially for few-shot settings.

**Weaknesses:**

-	The idea of using adaptive filers is interesting. However, the fitters still rely on some pre-defined thresholds, e.g., [0.005, 0.01, 0.05]. In practice, the authors may also need to tune these hyper-parameters to achieve the optimal performance for different datasets.
-	For CNN, according to Table 4 in Sec. B.2, the proposed method does not lead to further gains compared with MFM when it comes to full fine-tuning, which makes me concerned about its effectiveness for CNN architectures. Could the authors provide the justification on this?
-	For efficiency analysis, the authors only provide a comparison on GPU memory usage (Table 12, Sec. B.6). A comparison on training time with previous methods is also preferred.

**Questions:**

See the questions mentioned above. Overall, I think it is an interesting paper with extensive experiments and analysis, which could provide some new insights for the community. Thus, I am leaning to accept this paper.

---

> ### Author Response · Authors · 2024-11-20
>
> > Overall
>
> Thank you very much for your positive and encouraging feedback on our paper. We are glad to hear that you found our method “well-motivated”, the paper “well-written”, and that our proposed solutions effectively address the key limitations of MFM. Your support means a lot to us, and we hope our work provides valuable insights to the community.
>
> We have carefully addressed your concerns below. For one of your points, we would like to kindly direct you to our response to another reviewer, as this concern is shared between both reviews.
>
> > W1
>
> Thank you for your insightful observation regarding the reliance on pre-defined thresholds in our adaptive filters. We agree that hyperparameter tuning can be a concern when applying the method to different datasets. To address this, we have provided guidance in the paper (line 269 in the main text, and Appendix Section B.5.1) on selecting threshold values without extensive tuning.
>
> In practice, we found that selecting thresholds that retain or mask approximately 0.2% to 7% of the highest-magnitude frequencies yields good performance across various datasets. This range ensures that the frequency-masked images are visually distinct, enhancing the diversity of the training data. By observing the images produced with these thresholds, practitioners can balance retaining important image structures with introducing sufficient masking to challenge the model during pre-training. Please also see the images in Tables 13, 14, and 15 in the appendix that illustrate the effect of frequency masking with different thresholds.
>
> **Additionally, our empirical results demonstrate that the model's performance remains robust as long as the thresholds are spread out to produce varying levels of frequency masking (Appendix B.5.1 and Table 8).** This approach minimizes the need for fine-grained tuning of threshold values, simplifying the practical application of our method across different datasets.
>
> > W2
>
> Thank you for pointing out your concern regarding the effectiveness of FOLK with CNN architectures in the full fine-tuning setting. We acknowledge that, as shown in Table 4 of Section B.2, FOLK does not exhibit significant gains over MFM for CNN-based models when fully fine-tuned. However, we'd like to emphasize that one of FOLK's key strengths lies in few-shot learning scenarios, which are particularly critical in the self-supervised learning domain due to the common scarcity of labeled data.
>
> **In lines 961 to 999 in the appendix, and as presented in Table 5, we demonstrate that FOLK significantly outperforms MFM in few-shot settings, even with CNN backbones. Specifically, FOLK achieves substantially higher accuracy with only 1% and 10% of labeled data compared to MFM.** This highlights FOLK's robustness and effectiveness in scenarios with limited labeled data, making it a superior choice for practical applications where acquiring large amounts of labeled data is challenging.
>
> > W3
>
> Please see our comment on Weakness 1 (W1) from reviewer tJxA.

---

> ### Author Response · Authors · 2024-11-24
>
> Dear Reviewer 6EN5,
>
> We appreciate your valuable comments on our paper. We have prepared a rebuttal with an updated manuscript and tried our best to address most, if not all, of your concerns. We notice that the author-reviewer discussion period is coming to an end, and we are willing to answer any unresolved or further questions that you may have regarding our rebuttal if time is allowed.
>
> Thank you once again for your time and your encouraging feedback on our paper.
>
> Best,
>
> Authors

---

> > ### Comment · Reviewer_6EN5 · 2024-11-26
> > **Official Comment by Reviewer 6EN5**
> >
> > Thanks for the authors' rebuttal, which has addressed my concerns. I do not have further comments. I would like to keep my original rating.

---

> > > ### Author Response · Authors · 2024-11-27
> > >
> > > Thank you for reviewing our paper and for your thoughtful feedback. We're glad our rebuttal addressed most of your concerns. We appreciate your decision to support our work with the score.

---

### Official Review · Reviewer_DqRd · 2024-11-04

**Soundness:** 3
**Presentation:** 2
**Contribution:** 2
**Rating:** 6
**Confidence:** 3

**Summary:**

The paper introduces a self-supervised learning (SSL) approach named FOLK, which stands for FOurier transform compression with seLf-Knowledge distillation. The method aims to address the limitations of previous frequency-based pre-training approaches by adaptively selecting frequencies for masking based on unique image responses. The dual-branch framework leverages both filtered and original images during pre-training, which is claimed to minimize the adaptation requirements for natural-looking images in downstream tasks. The experimental results demonstrate the effectiveness of FOLK, showing competitive performance in various downstream tasks such as image classification, few-shot learning, and semantic segmentation.

**Strengths:**

The paper presents a new method that combines frequency-based masking with self-knowledge distillation, addressing known limitations in the field of SSL for computer vision tasks. The paper provides extensive experimental results that demonstrate FOLK's effectiveness across a range of tasks and benchmarks, showing improvements over existing state-of-the-art methods.

**Weaknesses:**

The author proposed two limitations in the introduction, but the experiments did not directly discuss how to address these limitations. Simply showing performance improvements (e.g., image classification tasks) is not enough to support the author's claims.

**Questions:**

1. The paper primarily focuses on the Com and RCom filters. It would be beneficial to see a comparison with other filtering techniques to establish the robustness and generalizability of the FOLK framework. Could the authors experiment with additional filtering methods, such as Gabor filters or wavelet transforms, and report on their effectiveness?
2. The related work section mentions that MFM has been applied to low-level vision tasks. Since FOLK builds upon MFM, it would be valuable to include a comparison of FOLK's performance on such tasks. Could the authors add experiments that benchmark FOLK against existing methods on low-level vision tasks to provide a more comprehensive evaluation?
3. While the supplementary material shows some results on robustness, a more detailed analysis would be appreciated. Could the authors provide additional benchmarks that specifically measure the robustness of the FOLK framework against various types of image degradations and noise?
4. The paper integrates knowledge distillation into the FOLK framework, but ablation studies regarding the contribution of this component are missing. Could the authors conduct ablation studies to isolate the impact of the knowledge distillation component on the overall performance? This would help readers understand the significance of this technique in the context of the proposed framework.

---

> ### Author Response · Authors · 2024-11-20
>
> > Overall
>
> Thank you for your thoughtful review and for acknowledging the effectiveness of our proposed method. We have carefully addressed your concerns below.
>
> > W1
>
> Thank you for pointing out that our experiments did not directly address the two limitations proposed in the introduction. We apologize for this oversight and appreciate the opportunity to clarify how our work addresses these issues.
>
> **Addressing the First Limitation:**
>
> To demonstrate our filters are effective, we show that they present a harder pretext task for the model during pretraining compared to the constant filters originally used in MFM, in terms of the loss progression. The results are as follows:
>
> | Epoch | MFM + Com/RCom Loss | MFM Loss |
> |:----------:|:----------:|:----------:|
> | 0 | 0.2878 | 0.2863 |
> | 19 | 0.1895 | 0.1811 |
> | 39 | 0.1821 | 0.1744 |
> | 59 | 0.1804 | 0.1736 |
> | 79 | 0.1789 | 0.1722 |
> | 99 | 0.1788 | 0.1718 |
>
> The higher loss values for MFM + Com/RCom indicate that our informed filters present a more challenging task for the model compared to the constant filters used in MFM. A higher loss suggests that the model finds it more difficult to predict the masked frequencies, which implies that our filters are more effective in encouraging the model to learn richer representations. This directly addresses the first limitation by providing a more complex and informative masking strategy.
>
> We have now included this analysis in Appendix Section B.5.5 in the updated manuscript.
>
> **Addressing the Second Limitation:**
>
> Regarding the second limitation, we would like to highlight the significant improvements achieved by our FOLK model in few-shot learning scenarios compared to MFM. While FOLK offers only marginal performance improvements over MFM in the full fine-tuning scenario, these gains are still meaningful. More importantly, in few-shot learning scenarios, FOLK demonstrates significantly greater enhancements, which more accurately reflect the limitations of MFM. As MFM lacks a mechanism to expose the model to natural images during pretraining, its performance in few-shot learning is greatly hampered—a limitation that may remain unnoticed in full fine-tuning due to the abundance of training data. In contrast, FOLK incorporates a self-distillation design that enables the student model to be exposed to natural, unmasked images by learning from the teacher model during pre-training. This approach mitigates MFM’s limitation and enhances the model’s adaptability and performance when labeled data is scarce.
>
> We appreciate your insightful feedback and hope that this clarification demonstrates how our experiments support the claims made regarding the proposed limitations.
>
> > Q1
>
> In our paper, we focused on the Com and RCom filters because they are specifically designed to highlight the main semantics of an image by preserving significant frequencies (Com) and reconstructing finer details (RCom). We discuss the rationale for choosing these filters in Section 3.2.1 (lines 255–264).
>
> To address your concern, we would like to kindly refer you to our experiments with additional filtering methods, including Gabor filters and other techniques. The results of these comparisons are detailed in Appendix B.5.4. Our ablation studies demonstrate that the Com/RCom filters outperform other filters in terms of model accuracy. For instance, as shown in Table 11, the Com/RCom filters achieved higher accuracy compared to Gabor filters, token masks, and circle masks when using a ViT-S/16 model. We believe this demonstrates the robustness and generalizability of our approach.
>
> We appreciate your recommendation and will consider exploring additional filtering methods, such as wavelet transforms, in future work to further validate and enhance the FOLK framework.
>
> > Q2
>
> Thank you for highlighting the importance of evaluating our proposed method on low-level vision tasks for a comprehensive assessment. We fully agree with your suggestion. In our paper, we have conducted experiments on semantic segmentation, which is a representative low-level vision task and the only one evaluated by MFM. Specifically, we evaluated FOLK on the ADE20K dataset, following the methodology used by MFM. The results of these experiments are detailed in Appendix B.1. Our findings show that FOLK outperforms MFM and other baseline methods on semantic segmentation, demonstrating that it not only excels in high-level vision tasks but also provides improved performance on low-level vision tasks.

---

> > ### Author Response · Authors · 2024-11-20
> >
> > > Q3
> >
> > Thank you for your insightful feedback regarding the robustness of the FOLK framework against image degradations and noise.
> > To address your concern, we kindly refer you to the Appendix Section B.3, where we assessed the impact of common types of noise on the pre-trained model via our FOLK framework, such as salt-and-pepper noise and Gaussian noise. To provide a more comprehensive analysis, we further conducted experiments to evaluate how our pre-trained model handles other types of image degradations, focusing on brightness and contrast variations.
> >
> > **Experimental Setup:**
> >
> > We randomly selected 100 images from the ImageNet-1k validation set and applied the following transformations to each image:
> >
> > - **Brightness Variation**: Adjusted the brightness by a random factor within ±50% to simulate different lighting conditions.
> > - **Contrast Variation**: Adjusted the contrast by a random factor within ±50% to mimic variations in image quality.
> >
> > For each original and transformed image, we applied the Com and RCom filters (as described in Section 3.2.1) to create frequency-masked versions. We then used our FOLK pre-trained model to predict the masked frequencies and calculated the MFM loss for each case.
> >
> > | Type | No Transformation | Brightness Variation |  Contrast Variation |
> > |:---------:|:---------:|:---------:|:---------:|
> > | MFM Loss | 0.179 | 0.186 | 0.189 |
> >
> > All results were obtained using a ViT-S/16 model. Full results can be found in the updated manuscript, Appendix Section B.3, including noisy cases that were originally presented in the paper.
> >
> > **Discussion:**
> >
> > The results indicate that introducing brightness and contrast variations leads to only a slight increase in the MFM loss—approximately 0.006 to 0.010 higher than that of the original images. This minimal difference suggests that our model's ability to reconstruct masked frequencies remains robust even under significant changes in brightness and contrast.
> >
> > > Q4
> >
> > We fully agree with your suggestion that isolating the impact of the knowledge distillation component is essential to help readers better understand its significance. In our paper, we have included comparative experiments that effectively serve as ablation studies for this purpose. We would like to kindly direct your attention to the following items in our results tables:
> >
> > - “MFM*”: Our reproduced results with MFM official code.
> > - “MFM + R/Com*”: Replacing the constant filters in MFM with our proposed Com/Rcom filters, and without the knowledge distillation design.
> > - ”FOLK”: The full FOLK framework with both Com/Rcom filters and knowledge distillation.
> >
> > We believe that this experimental setup facilitates a direct analysis of the contributions of each of our proposed components. Specifically, by comparing “MFM + R/Com*” with “MFM*”, we observe the improvements offered by our proposed filters; by comparing “FOLK” with “MFM + R/Com*”, we observe the enhancements resulting from our knowledge distillation design.
> > These three setups have been introduced in Table 1 (lines 449–452), and have been evaluated in all of our experiments, presented in Tables 1, 2, 3, 4, 5, and 7. The consistent improvement in performance across various tasks and benchmarks highlights the contribution of the knowledge distillation component to the overall effectiveness of our framework.

---

> > > ### Comment · Reviewer_DqRd · 2024-12-02
> > >
> > > Thank you for the detailed response. I still have some concerns that have not been addressed.
> > >
> > > * In Table 1, there is a noticeable difference between the reproduced results of MFM and the original results of MFM. Please explain the reasons for this difference. Compared with MFM's original results, it seems that FOLK does not have a significant advantage at 300 epochs. Then, the contribution of FOLK may come more from more training (i.e., 800 epochs).
> > >
> > > * The authors still lack detailed results in the robustness evaluation. Please refer to MFM's section 4.5 for assessment results of adversarial robustness, common corruption robustness, and out-of-distribution robustness. It is not sufficiently effective to only show the impact on loss values with a few simple degradations.

---

> > > > ### Author Response · Authors · 2024-12-02
> > > >
> > > > Thank you for your continued feedback. We appreciate the opportunity to clarify our work.
> > > >
> > > > We understand that you have some NEW concerns that have not been raised with us previously until a day before the discussion is supposed to close. Specifically:
> > > >
> > > > 1. **Difference Between Reproduced and Original MFM Results:** We are happy to clarify this matter in the following comment.
> > > >
> > > > 2. **Robustness Evaluation:** Initially, you asked if we could provide additional benchmarks measuring the robustness of the FOLK framework against various types of image degradations and noise. In response, we conducted experiments focusing on image noise and artifacts. In your latest feedback, you are requesting significant experiments on robustness evaluation similar to MFM's assessments, including adversarial robustness, common corruption robustness, and out-of-distribution robustness. **We recognize the importance of these evaluations and appreciate your valuable suggestions. However, due to the timing of your request, we regret that we are unable to conduct these additional experiments within the given timeframe**. We ask for your kind consideration, given the circumstances. Nevertheless, the existing experiments that we have conducted on robustness should have shed some light on this critical aspect.

---

> > > > ### Author Response · Authors · 2024-12-02
> > > >
> > > > > **Q1**:
> > > >
> > > > 1. **Difference Between Reproduced and Original MFM Results:** We noticed slight discrepancies between our reproduced MFM results and those reported originally:
> > > >
> > > >     Original MFM at 300 Epochs:
> > > >     * ViT-S/16: 81.6%
> > > >     * ViT-B/16: 83.1%
> > > >
> > > >     Our Reproduced MFM at 300 Epochs:
> > > >     * ViT-S/16: 81.2%
> > > >     * ViT-B/16: 82.9%
> > > >
> > > >     These differences may be due to factors like random seed variations, hardware differences, hyperparameters, or minor implementation details. We reproduced these results using the official MFM code they published, as we mentioned in the paper. However, it seems there are differences between the hyperparameters reported in their paper and those in their official code, which could account for the discrepancies. We used the official MFM code without modifications to ensure a fair comparison under identical conditions.
> > > >
> > > > 2. **FOLK's Advantage at 300 Epochs:** Even when considering the original MFM results (**for having a fair comparison, we added both the original and our reproduced results in the table**), FOLK matches or slightly surpasses them:
> > > >
> > > >     * ViT-S/16: FOLK achieves 81.6%, matching MFM.
> > > >     * ViT-B/16: FOLK achieves 83.4%, a 0.3% improvement over MFM's 83.1%.
> > > >
> > > >     While the improvement is modest, it is meaningful given the scale of ImageNet-1K.
> > > >
> > > > 3. **Superior Performance in Few-Shot Learning:** FOLK's key advantage emerges in low-labeled data scenarios. In our few-shot learning experiments (Section 4.2.2), fine-tuning with only 10% of ImageNet-1K data:
> > > >
> > > >     FOLK at 300 Pre-training Epochs:
> > > >     * Achieves up to 71.2% accuracy.
> > > >     * **Outperforms MFM (up to 58.5%)** and other methods under various settings.
> > > >
> > > > > **Q2:**
> > > >
> > > > 1. **Clarification of Our Robustness Claims:**
> > > >
> > > >     We would like to clarify that in our work, we focused specifically on evaluating the robustness of image noise and artifacts, as detailed in Appendix Section B.3, titled "Robustness to Image Noise and Artifacts." Our intention was to assess how our **pre-trained model** (what you asked for is for the fine-tuned model) handles common image degradations that affect frequency components, given that our method leverages frequency masking during pre-training. **We aimed to address concerns specifically related to the potential sensitivity of the Fourier transform to image artifacts and noise (refer to lines 1012-1016)**. By focusing on this aspect, we provided evidence that our model is robust in reconstructing masked frequencies even when the input images contain typical degradations, such as salt-and-pepper noise, Gaussian noise, brightness variations, and contrast variations.
> > > >
> > > >     We did not claim that our **finetuned model** is robust to adversarial attacks, common corruptions, or out-of-distribution scenarios beyond the specific types of image noise and artifacts we tested.
> > > >
> > > > 2. **Future Directions:**
> > > >
> > > >     We appreciate your suggestion and recognize the importance of evaluating models under a wider range of robustness benchmarks. While we did not claim general robustness in this work, we agree that extending our evaluations to include:
> > > >
> > > >     * Adversarial Robustness
> > > >     * Common Corruption Robustness
> > > >     * Out-of-Distribution Robustness
> > > >
> > > >      would offer a more comprehensive understanding of our model's performance. We consider this an important direction for future research and plan to incorporate these assessments in subsequent studies.
> > > >
> > > > > **We would like to emphasize FOLK's significant advantage in the few-shot fine-tuning regime**.
> > > >
> > > > As demonstrated in our experiments (Section 4.2.2), FOLK outperforms existing methods, particularly when labeled data is limited—a critical aspect in self-supervised learning. This highlights the practical value of our approach in real-world scenarios where data annotation is costly or impractical.
> > > >
> > > > Thank you again for your valuable insights. Please let us know if you have further questions.

---

> > > > > ### Comment · Reviewer_DqRd · 2024-12-03
> > > > >
> > > > > Although the authors have not fully addressed my concerns, I believe they can further refine their experiments in the final version. Considering the opinions of other reviewers, I have decided to raise my score to a positive rating.

---

> ### Author Response · Authors · 2024-11-24
>
> Dear Reviewer DqRd,
>
> We appreciate your valuable comments on our paper. We have prepared a rebuttal with an updated manuscript and tried our best to address most, if not all, of your concerns. We notice that the author-reviewer discussion period is coming to an end, and we are willing to answer any unresolved or further questions that you may have regarding our rebuttal if time is allowed.
>
> If our rebuttal has addressed your concerns, we would appreciate it if you would be willing to consider raising your original rating. Thank you for your time and consideration.
>
> Best,
>
> Authors

---

### Official Review · Reviewer_nBwS · 2024-11-04

**Soundness:** 3
**Presentation:** 3
**Contribution:** 2
**Rating:** 6
**Confidence:** 3

**Summary:**

This paper introduces Fourier transform compression with self-knowledge distillation (FOLK), a frequency-based self-supervised learning (SSL) method designed to improve pre-training efficiency. FOLK addresses the limitations by adaptively selecting masked frequencies based on image frequency responses and employing a two-branch framework for knowledge distillation. Experimental results show that FOLK achieves competitive performance across various SSL tasks.

**Strengths:**

1. The framework is applicable and straightforward to understand.
2. The proposed method improves the learning of the student model and facilitates a more efficient training process.
3. The paper presents experiments across multiple datasets and various vision tasks, demonstrating the effectiveness of the proposed method.

**Weaknesses:**

1. The dual-stream and frequency-domain masking approaches applied in the article are relatively common schemes. Could the authors elaborate further on the motivation of the proposed method?
2. More analysis and experiments are required on the framework design and cost computation, please see the questions.

**Questions:**

1. Two views (u and v) of the input image are processed through the informed filtering process in the proposed FOLK framework. What is the optimal method for selecting views to enhance the model performance?
2. How can the complexity of the FOLK be reduced to enhance the framework's accessibility? Could you analyze the computational costs of the various methods evaluated across different models?

---

> ### Author Response · Authors · 2024-11-20
>
> > Overall
>
> Thank you for your thoughtful review and for highlighting the applicability and clarity of our framework. We have carefully addressed your concerns below.
>
> > W1
>
> Thank you for your insightful comment. While we agree with the reviewer that dual-stream and frequency-domain masking approaches are common, our motivation for FOLK is to address two specific limitations of Masked Frequency Modeling (MFM).
>
> Firstly, MFM lacks a mechanism to expose the model to natural views of images during pre-training. This limitation especially hampers performance in few-shot learning scenarios (see “MFM” in Table 2). To specifically address this limitation, we incorporated self-distillation into FOLK, as it perfectly serves to familiarize the model with natural images during pre-training. In our approach, the teacher model processes the original image (information-rich), while the student model processes the frequency-masked image (information-poor) and receives guidance from the teacher’s output. Though this is analogous to DINO’s [1] teacher global views (information-rich) and student local views (information-poor), FOLK’s specific self-distillation design has not been explicitly explored by previous studies, and it does not require as many student frequency-masked views as DINO’s local views (i.e. 6). Such an approach allows the student to benefit from both masked and natural image representations, enhancing training stability and improving downstream task performance. This has been elaborated in lines 107-114, 312-318, 388-401, and 1345-1352 in the paper.
>
> Secondly, though MFM offers unique advantages compared to Masked Image Modeling (lines 52-80), the constant frequency filters used in the original MFM result in training inefficiency (lines 248-253, and “MFM*” compared to “MFM + R/Com*” in Tables 1 and 2). To address this limitation, we propose the Com/Rcom filters to adaptively mask (in)significant frequencies based on each image’s unique frequency components, hence presenting a challenging pretext task for effective pre-training (lines 255-264, Tables 1 and 2, and Appendix Section B.5.5).
>
> [1] Caron, Mathilde, et al. "Emerging properties in self-supervised vision transformers." Proceedings of the IEEE/CVF international conference on computer vision. 2021.
>
> > W2
>
> Please see our comments on Q1 and Q2 below.
>
> > Q1
>
> Thank you for raising the question regarding the selection of views u and v in the FOLK framework. We agree that the data augmentation techniques used to create different views are critical factors for model training. Previous studies have extensively explored this topic, including iBOT [1], DINO [2], AttMask [3], and FixMatch [4]. In our work, we have followed the same augmentations used in DINO to facilitate a fair comparison with other methods.
>
> We apologize for not being clear on how we chose these augmentation steps, and we have now clarified it in the updated manuscript. Please refer to lines 321-322.
>
> [1] Zhou, Jinghao, et al. "ibot: Image bert pre-training with online tokenizer." arXiv preprint arXiv:2111.07832 (2021).
>
> [2] Caron, Mathilde, et al. "Emerging properties in self-supervised vision transformers." Proceedings of the IEEE/CVF international conference on computer vision. 2021.
>
> [3] Kakogeorgiou, Ioannis, et al. "What to hide from your students: Attention-guided masked image modeling." European Conference on Computer Vision. Cham: Springer Nature Switzerland, 2022.
>
> [4] Sohn, Kihyuk, et al. "Fixmatch: Simplifying semi-supervised learning with consistency and confidence." Advances in neural information processing systems 33 (2020): 596-608.
>
> > Q2
>
> Please see our comment on Weakness 1 (W1) from reviewer tJxA.

---

> ### Author Response · Authors · 2024-11-24
>
> Dear Reviewer nBwS,
>
> We appreciate your valuable comments on our paper. We have prepared a rebuttal with an updated manuscript and tried our best to address most, if not all, of your concerns. We notice that the author-reviewer discussion period is coming to an end, and we are willing to answer any unresolved or further questions that you may have regarding our rebuttal if time is allowed.
>
> If our rebuttal has addressed your concerns, we would appreciate it if you would be willing to consider raising your original rating. Thank you for your time and consideration.
>
> Best,
>
> Authors

---

> ### Comment · Reviewer_nBwS · 2024-12-03
>
> Thank you for answering my concerns.

---

### Official Review · Reviewer_tJxA · 2024-11-04

**Soundness:** 3
**Presentation:** 3
**Contribution:** 3
**Rating:** 6
**Confidence:** 5

**Summary:**

This paper proposes a frequency-based SSL method to learn visual representations from unlabeled images, which significantly improves the training performance compared with existing works. In particular, the authors built upon the MFM method and identified two key limitations: 1) pre-defined frequency masking filters that ignore the intrinsic structure in individual images; 2) model pre-trained with frequency-filtered images leads more data to adapt to natural images in downstream model fine-tuning. In response, two specific new designs (a. masked frequency modeling with Com and RCom filters; b. multi-task self-supervision with self-knowledge distillation) are proposed to target these two problems. Their reported experimental studies have shown the effectiveness of their designs.

**Strengths:**

**Originality**. The paper investigated two fundamental limitations in the MFM work and proposed two novel designs to address these limitations.  The presentation clearly shows what are the novel elements.

**Quality**.  The paper shows a successful way to perform masking in the frequency domain for unlabeled training images. Additionally, the authors provided a proper self-knowledge distillation framework to deal with the negative effect of training with frequency-masked images.

**Clarity**.  The submission neatly shows all the experiments that were carried out and the description of the underlying method is clear.

**Significance and Relevance**.  The topic is very interesting and important. Considering the growing demand for learning effective representations from unlabeled data, this paper pushed the boundary of SSL.

**Weaknesses:**

**Training Cost**. Given that the proposed method employs a two-branch framework for model training, will it bring additional training costs compared with the original MFM?

**Masking Filters**.  What are the exact formulations of Com and RCom masking? or pseudo code to construct Com and RCom might be helpful.

**Data Augmentations**. In generating two views, u and v, distinct transformations (random cropping, color jittering, etc.) are conducted. It seems no ablation studies are provided for analyzing the effect on the consecutive image frequency masking and final model training.

**Questions:**

See above weaknesses.

---

> ### Author Response · Authors · 2024-11-20
>
> > Overall
>
> Thank you for your positive feedback. We are pleased to hear that you found our work "novel" and our presentation "clear." We appreciate your recognition of our contributions to the field of self-supervised learning.
>
> We have carefully addressed your concerns below. Regarding one of your points, we kindly refer you to our response to another reviewer, as this concern overlaps with theirs.
> Answers:
>
> >W1
>
> Thank you for highlighting this important consideration regarding the training cost of our proposed method. We acknowledge that our dual-branch framework, which incorporates knowledge distillation, introduces additional computational overhead compared to the original MFM. This increased cost stems from processing both the teacher and student models during pre-training.
>
> However, we designed this framework specifically to address key limitations of MFM, particularly its inability to effectively handle few-shot learning scenarios. By enabling the model to learn from both masked and unmasked views of the data through self-distillation, we achieve significant improvements across various downstream tasks, especially in few-shot learning scenarios. We believe that these improvements justify the additional computational resources required during pre-training.
>
> **Moreover, it's important to note that pre-training is a one-time investment. Once the model is pre-trained, it can be fine-tuned for various tasks without incurring the same level of computational cost again.** In many practical applications, the enhanced performance and versatility of the pre-trained model can outweigh the initial training overhead. The following table provides a detailed comparison between FOLK and MFM in terms of training time (number of seconds per epoch on the ImageNet-1K dataset), GPU usage, and few-shot learning accuracy (from Table 2 in the paper).
>
> | Model | Time (s/epoch) | GPU (GB) | Few-shot AVG Accuracy (%) | Few-shot MAX Accuracy (%) |
> |:---------:|:---------:|:---------:|:---------:|:---------:|
> | MFM | 2083 | 10.1 | 52.7 | 58.5 |
> | FOLK | 4965 | 19.8 | 67.2 | 71.2 |
>
> >W2
>
> Thank you for highlighting the need for more explicit formulations of the Com and RCom masking filters. While Section 3.2.1 (“Informed Filters”) in the main paper discusses the conceptual design of these filters, we agree that providing exact equations or pseudocode would greatly enhance the clarity and facilitate a better understanding of our method for the readers.
>
> We have now added pseudocode to the appendix (Section A.4) to better illustrate the implementation details of the proposed Com and Rcom filters.
>
> >W3
>
> Please see our comment on Question 1 (Q1) of reviewer nBwS.

---

> ### Author Response · Authors · 2024-11-24
>
> Dear Reviewer tJxA,
>
> We appreciate your valuable comments on our paper. We have prepared a rebuttal with an updated manuscript and tried our best to address most, if not all, of your concerns. We notice that the author-reviewer discussion period is coming to an end, and we are willing to answer any unresolved or further questions that you may have regarding our rebuttal if time is allowed.
>
> If our rebuttal has addressed your concerns, we would appreciate it if you would be willing to consider raising your original rating. Thank you for your time and consideration.
>
> Best,
>
> Authors

---

> > ### Comment · Reviewer_tJxA · 2024-11-26
> > **Rebuttal Acknowledgment**
> >
> > I thank the reviewers for their detailed clarification, which has addressed my concerns about their work. Therefore, I keep my rating as accepted.

---

> > > ### Author Response · Authors · 2024-11-27
> > >
> > > Thank you for taking the time to review our paper, provide constructive feedback, and support it with an accepted score. Your suggestions have been invaluable in improving our work.

---

### Author Response · Authors · 2024-11-20

We thank the reviewers for their tremendous efforts in the paper review process. We are glad that the reviewers found our proposed method “novel” (tJxA) and “well motivated” (6EN5), our experiments “extensive” (6EN5, DqRd), and our paper “well-written” (6EN5) and “straightforward to understand” (nBwS). We appreciate all the review comments and have tried our best to address most, if not all, of the concerns.

According to the reviewers' suggestions, we have now made modifications to the manuscript to better clarify our method. **Modifications are highlighted in red texts in the updated manuscript for easy identification. All line numbers mentioned in our responses are according to the modified version of the paper.**

---

### Author Response · Authors · 2024-11-21

Dear Reviewers,

Thank you for taking the time to review our paper. We appreciate the insightful questions and feedback provided in your initial review. We have carefully considered your comments and have addressed each point in our rebuttal. However, we are more than willing to engage in further discussion if you have any further questions, require clarification on any aspect of our work, or wish to suggest additional improvements.

Thank you again for your valuable feedback, and we look forward to hearing from you. Best regards, Authors

---

### Meta-Review · Area_Chair_aRrm · 2024-12-17

**Metareview:**

The paper introduces a frequency-based self-supervised learning method that improves pre-training efficacy by addressing limitations in masked frequency modeling (MFM). The proposed approach adaptively masks image frequencies and employs self-knowledge distillation. Reviewers appreciated the novelty, motivation, and extensive experiments, particularly highlighting improvements in few-shot learning. Concerns were raised about training cost, hyperparameter tuning, and the dual-branch complexity, but the authors’ clarifications and modifications addressed most of these. Given the overall positive feedback, the AC recommends acceptance.

**Additional Comments On Reviewer Discussion:**

Reviewers raised concerns about training cost (tJxA), filter effectiveness (6EN5, DqRd), and framework complexity (nBwS). The authors addressed these by clarifying computational trade-offs, providing pseudocode for filters, and discussing performance gains, particularly in few-shot learning. Additional robustness tests were requested but deemed impractical within the timeframe. The authors promised to add those experiments in in the future version. Given the well-motivated solution and extensive experiments, most reviewers increased their scores. Balancing concerns and strengths, the paper’s contributions justified acceptance.

---

### Decision · Program_Chairs · 2025-01-22

Accept (Poster)